# Development of A4 antibody for detection of neuraminidase I223R/H275Y-associated antiviral multidrug-resistant influenza virus

Kyeonghye Guk[1,2], Hyeran Kim[1], Miyeon Lee[3], Yoon-Aa Choi[4], Seul Gee Hwang[1,2], Gaon Han[1,2], Hye-Nan Kim[1], Hongki Kim[1], Hwangseo Park[5], Dongeun Yong[6], Taejoon Kang ● [1✉], Eun-Kyung Lim[1,2✉] & Juyeon Jung ● [1,2✉]

The emergence and spread of antiviral drug-resistant viruses have been a worldwide challenge and a great concern for patient care. We report A4 antibody specifically recognizing and binding to the mutant I223R/H275Y neuraminidase and prove the applicability of A4 antibody for direct detection of antiviral multidrug-resistant viruses in various sensing platforms, including naked-eye detection, surface-enhanced Raman scattering-based immunoassay, and lateral flow system. The development of the A4 antibody enables fast, simple, and reliable point-of-care assays of antiviral multidrug-resistant influenza viruses. In addition to current influenza virus infection testing methods that do not provide information on the antiviral drug-resistance of the virus, diagnostic tests for antiviral multidrug-resistant viruses will improve clinical judgment in the treatment of influenza virus infections, avoid the unnecessary prescription of ineffective drugs, and improve current therapies.

[1] Bionanotechnology Research Center, Korea Research Institute of Bioscience & Biotechnology (KRIBB), 125 Gwahak-ro, Yuseong-gu, Daejeon 34141, Republic of Korea. [2] Department of Nanobiotechnology, KRIBB School of Biotechnology, University of Science and Technology (UST), 217 Gajeong-ro, Yuseong-gu, Daejeon 34113, Republic of Korea. [3] Department of Chemistry, Korea Advanced Institute of Science and Technology (KAIST), 291 Daehak-ro, Yuseong-gu, Daejeon 34141, Republic of Korea. [4] BioNano Health Guard Research Center, KRIBB, 125 Gwahak-ro, Yuseong-gu, Daejeon 34141, Republic of Korea. [5] Department of Bioscience and Biotechnology, Sejong University, 209 Neungdong-ro, Kwangjin-gu, Seoul 05006, Republic of Korea. [6] Department of Laboratory Medicine and Research Institute of Bacterial Resistance, Yonsei University College of Medicine, Seoul 03722, Republic of Korea. ✉email: kangtaejoon@kribb.re.kr; eklim1112@kribb.re.kr; jjung@kribb.re.kr

nfluenza is the leading cause of serious respiratory infections, which can lead to hospitalization and death. Traditionally, two types of antiviral drugs have been used in therapy: adamantines and neuraminidase inhibitors (NAIs). However, adamantine resistance among circulating influenza A viruses, including the A/H1N1 2009 pandemic influenza virus, is already prevalent worldwide due to increased selection pressure in the presence of drug treatment. Thus NAIs remain the only antiviral drugs (e.g., zanamivir[1] and oseltamivir[2]) used in current clinical settings[3]. Owing to its oral bioavailability and easy administration, oseltamivir has been the first choice for treating A/H1N1 2009 pandemic influenza virus infection and for inclusion in pandemic antiviral stockpiles[4]. Unfortunately, the emergence of oseltamivir-resistant variants, especially among A H1N1 viruses, has been reported and has become a great concern for patient care. In early clinical trials in the UK, resistance after oseltamivir treatment has been reported in up to 27% of children infected with A H1N1 viruses[5,6]. Moreover, approximately 25% of H5N1-infected patients develop resistance after oseltamivir treatment, leading to fatalities[7]. Early optimism was eventually dispelled by the emergence of oseltamivir-resistant A H1N1 viruses even in the absence of drug use and the spread of this resistance to the epidemic A H1N1 viruses[8–10].

In 2010, an isoleucine-to-arginine change at position 223 (I223R, N1 numbering) in the neuraminidase (NA) of A H1N1 virus isolated from an immunosuppressed child on prolonged oseltamivir and zanamivir therapy was reported[11]. In contrast to the frequently observed H275Y (N1 numbering) change, which causes selective resistance to oseltamivir[12,13], the I223R mutation confers a phenotype resistant to both oseltamivir and zanamivir[14,15]. In clinical cases, the I223R change has been found mostly in combination with H275Y. This double mutation (I223R/H275Y) in the NA of the 2009 pandemic H1N1 influenza virus has been reported as an antiviral multidrug-resistant virus for both zanamivir and oseltamivir[11,16–19]. In animal models and in vitro studies, I223R/H275Y double-mutant virus has shown high levels of resistance to oseltamivir and zanamivir, and the combination of I223R with H275Y does not compromise the replication capacity or transmissibility of the virus[16,20]. Considering that the oseltamivir and zanamivir is the two most widely used drugs for the treatment of influenza virus, the diagnosis of double-mutant viruses is critical[21].

Commercially available rapid influenza diagnostic tests (RIDTs) are immunoassays that utilize influenza virus-specific antibody affinity for the presence of the influenza virus antigens[22]. Although RIDT is easy to use, cost-effective, and can efficiently diagnose and treat influenza viruses within short processing times (minutes), RIDTs currently in use do not provide information on the antiviral drug resistance of influenza viruses, because antiviral drug-resistant influenza virus-specific antibodies have not yet been developed. For the identification of antiviral-resistant viruses, conventional plaque reduction assays[23,24], DNA sequencing of genes, and real-time polymerase chain reaction (RT-PCR) have been used. However, these methods are time consuming, labor intensive, expensive, and require highly trained personnel[25–28]. In order to diagnose drug-resistant viruses in immunoassays, it is priority to secure antibodies with high affinity for binding to mutant viruses.

Here we develop a monoclonal antibody, A4, specific to I223R/H275Y NA. According to the experimental and simulation results, the binding affinity of A4 for oseltamivir- and zanamivir-resistant mutant I223R/H275Y NA is approximately 600 times stronger than its binding affinity for oseltamivir- and zanamivir-susceptible wild-type (wt) NA. This difference in the binding affinity of the A4 antibody for I223R/H275Y NA and wt NA allows to develop several detection methods for antiviral

multidrug-resistant influenza viruses. First, the A4 antibody is tested on a naked-eye detection using A4-gold nanoparticle (Au NP) aggregation. Only in the presence of I223R/H275Y pH1N1, color of A4-Au NPs changes from red to purple. Second, we apply A4 antibody to the surface-enhanced Raman scattering (SERS)-based immunoassay for I223R/H275Y viruses. The combination of A4 antibody with SERS improves the detection limit of I223R/H275Y virus to 1.5 plaque-forming units (PFU). Third, A4-based lateral flow immunoassay (LFA) system is developed for the rapid diagnosis of I223R/H275Y pH1N1. The developed LFA system can identify I223R/H275Y pH1N1 even in the mixture with wt pH1N1. Furthermore, we successfully detect the mutant viruses in nasopharyngeal samples from patients, suggesting the practical applicability of the A4-based LFA for the diagnosis of multidrug-resistant influenza virus-infected patients. We anticipate that the A4 antibody-based, rapid, simple, and efficient detection of antiviral multidrug-resistant influenza viruses will prevent both the unnecessary administration of ineffective drugs and the spread of antiviral-resistant viruses due to continuous drug administration.

## Results

**Production of A4 antibody**. For the development of A4 antibody specific to I223R/H275Y NA, we first prepared influenza virus subtype H1N1 NA proteins, i.e., drug-susceptible wt NA and oseltamivir- and zanamivir-resistant mutant I223R/H275Y NA. These NAs were expressed in *Spodoptera frugiperda* (Sf9) cells using a baculovirus expression system, purified, and confirmed by sodium dodecyl sulfate polyacrylamide gel electrophoresis (SDS-PAGE) (Supplementary Fig. 1A) and western blot analysis (Supplementary Fig. 1B). Previously, it has been reported that I223R/H275Y NA exhibits an overall loose structure with disturbed positions but with a local rearrangement of the compact array at the drug-binding site[19]. The $K_I$ of I223R/H275Y NA for oseltamivir was significantly increased by 7500 times compared to that of wt NA due to the increase in the dissociation rate constant ($k_{off}$) for oseltamivir (15-fold)[19]. The $K_I$ of I223R/H275Y NA for zanamivir was 22-fold greater than that of wt NA due to the reduced association rate constant ($k_{on}$) and increased $k_{off}$[18].

A monoclonal A4 antibody specifically recognizing I223R/H275Y NA over wt NA was developed using a biopanning protocol and a phage display library of fragment antigen bindings (Fabs) against I223R/H275Y NA. Each round of panning included a subtraction step against wt NA followed by panning against I223R/H275Y NA to remove wt NA binders. After the third and fourth rounds of panning, the enriched pool was screened for I223R/H275Y NA-specific binders (Fig. 1a). Randomly selected clones were subjected to soluble Fab expression and screened by enzyme-linked immunosorbent assay (ELISA) for their binding affinity to I223R/H275Y NA. Among them, clone strongly bound to I223R/H275Y NA was selected and subjected to DNA sequencing, which confirmed the presence of unique clone (Fig. 1b). Immunoglobulin G (IgG) was converted from the Fab form to the whole IgG form for further application (Fig. 1c). IgG bivalency leads to binding affinities much higher than those of Fabs through avidity effects[29].

**Binding of A4 antibody to I223R/H275Y NA protein**. The binding affinities of the purified whole antibodies to I223R/H275Y NA protein were measured by ELISA. The selected A4 antibody bound to I223R/H275Y NA in a concentration-dependent manner with $K_d$ of 3.50 nM (Supplementary Fig. 2A). $K_d$ of A4 to wt NA was 2.03 μM (Supplementary Fig. 2B), demonstrating the approximately 600-fold tighter binding of A4 to I223R/H275Y NA than to wt NA. The affinity and kinetics of the A4 binding

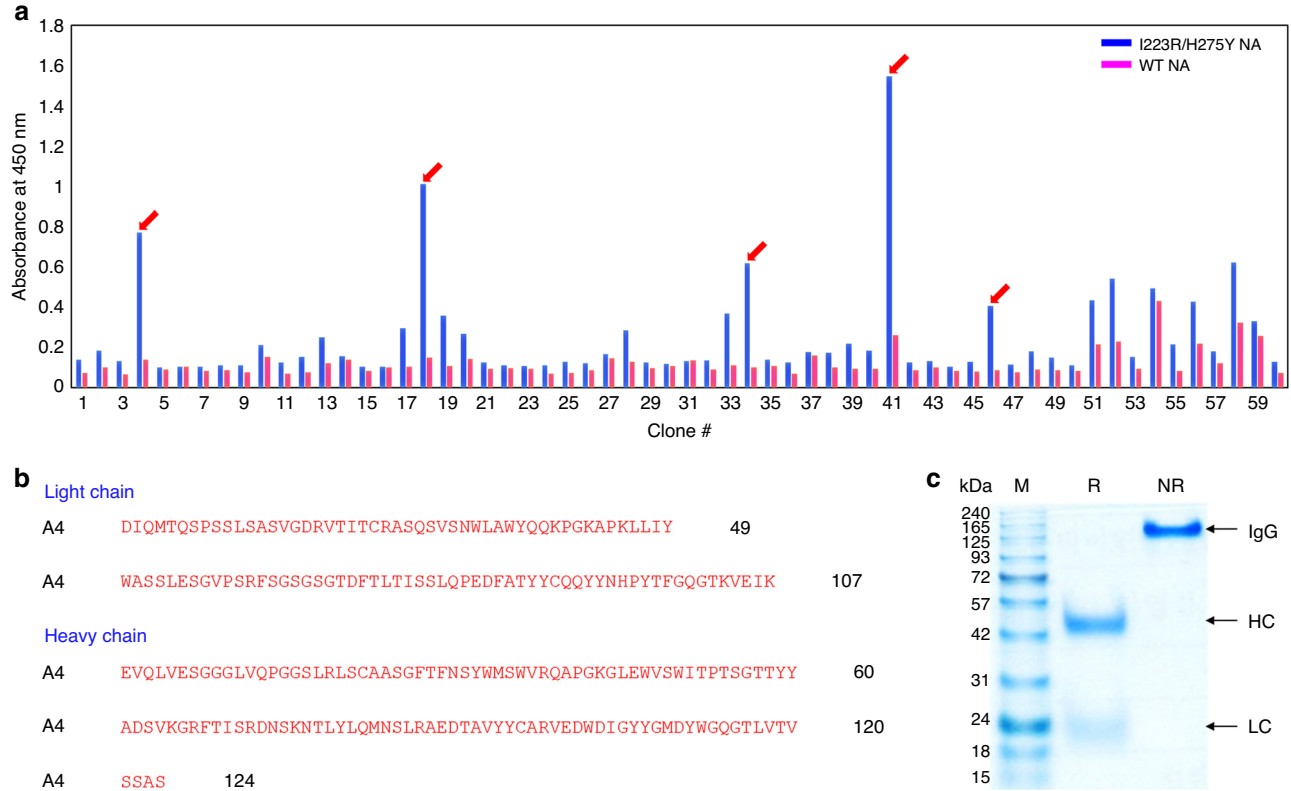

**Fig. 1 Selection of I223R/H275Y NA-specific monoclonal antibody. a** Screening for the specificity of monoclonal phages from third round of panning by ELISA. Each experiment was repeated two times independently with similar results. **b** Amino acid sequence of A4. **c** SDS-PAGE of purified A4 under nonreducing (NR) and reducing (R) conditions. Each experiment was repeated three times independently with similar results. Lane M: protein marker; HC: heavy chain, LC: light chain.

interaction to I223R/H275Y NA was also determined by surface plasmon resonance (SPR) (Fig. 2a). A4 was tested over a concentration range of 31.25–2000 nM on a low-density antigen-coated chip. The $k_{on}$ and $k_{off}$ of A4 to I223R/H275Y NA were $2.84 \times 10^3 \, M^{-1} \, s^{-1}$ and $7.22 \times 10^{-5} \, s^{-1}$, respectively; thus steady state affinity ($K_D$) was determined to be 0.254 nM. A4 was further characterized for its binding specificity to I223R/H275Y NA compared to wt NA using a dot-blot test by spotting different amounts of I223R/H275Y and wt NA proteins on a nitrocellulose (NC) membrane. The intensity of the dots corresponding to the binding interaction between I223R/H275Y NA and A4 was significantly stronger than that between wt NA and A4 (Fig. 2b, c).

In addition, we tested the recognition of single-mutant influenza NA protein (H275Y NA) using A4 antibody. H275Y mutation is the most frequently observed drug-resistant mutation[10]. The A4 antibody bound to H275Y NA in a concentration-dependent manner with $K_d$ of 0.12 μM (Supplementary Fig. 3A). Supplementary Fig. 3B displays the interaction between A4 antibody and pH1N1/H275Y mutant virus ($10^7$ PFU mL$^{-1}$) by dot-blot analysis. For the comparison, I223R/H275Y pH1N1 ($10^7$ PFU mL$^{-1}$) and wt NA were also examined. As shown in Supplementary Fig. 3B, the dot was observable only from the double-mutant virus. This suggests the low affinity of A4 antibody to the single-mutant influenza virus.

**Structural feature of A4 antibody and NA epitope.** Although three-dimensional (3D) structure of A4 has not been solved yet, we found the G6 antibody specific for binding to vascular endothelial growth factor (VEGF) as the structural template of A4 because the sequence identity amounts to 85.6% (Supplementary

Fig. 4). It has been reported that the root-mean-square deviation between the experimental and the predicted structures has fell within 1.5 Å when the sequence identity between the template and the target exceeds 50%[30]. Figure 3a, b are the comparative view of the docking poses of A4 in the wt NA and I223R/H275Y NA. The calculated binding modes of the epitope models for wt NA and I223R/H275Y NA reveal the different patterns for complexation. The binding free energy of the A4–I223R/H275Y NA complex is 1.8 kcal mol$^{-1}$ lower than that of the A4–wt NA complex, which is in agreement with the higher binding affinity of the former than the latter. The wt NA appears to be stabilized in complementarity-determining region (CDR) of A4 through the hydrogen bonds with the backbone aminocarbonyl atom of Asp101 in the heavy chain and with the side-chain carboxylate group of Asp103 in the heavy chain (Fig. 3c). On the other hand, the side-chain carboxylate ion of Glu100 and the two backbone amidic moieties of Asn30 and Ser31 in the heavy chain are involved in the hydrogen-bond stabilization of I223R/H275Y NA in CDR of A4 (Fig. 3d). Judging from the structural features derived with docking simulations, the higher binding affinity of I223R/H275Y NA to A4 than the wt NA may be attributed in a large part to the strengthening of the hydrogen-bond interactions in the antibody–epitope complex. Remarkably, the side chain of Leu224 of the wt NA is accommodated in a small hydrophobic pocket comprising the light and heavy chains of CDR. As a consequence of the double mutations, on the other hand, the side-chain phenolic moiety of Tyr275 binds to the hydrophobic pocket in the A4–I223R/H275Y NA complex instead of Leu224. This van der Waals contact seems to be stronger than that in the A4–wt NA complex because aromatic–aromatic hydrophobic interactions are stronger in general than aliphatic–aromatic ones.

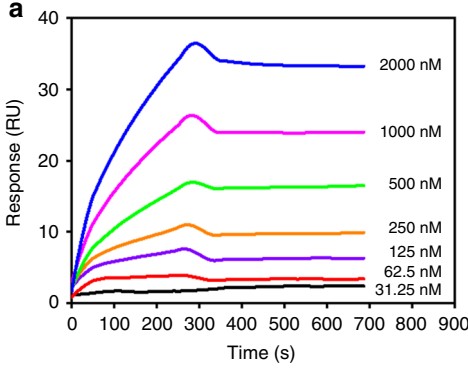

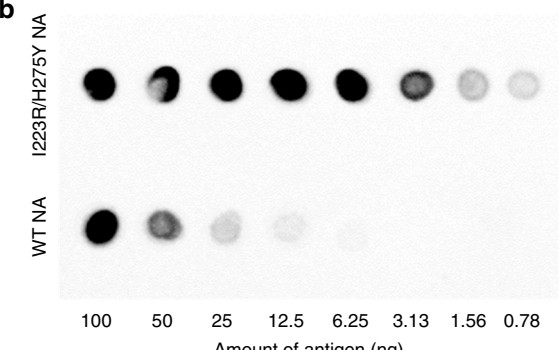

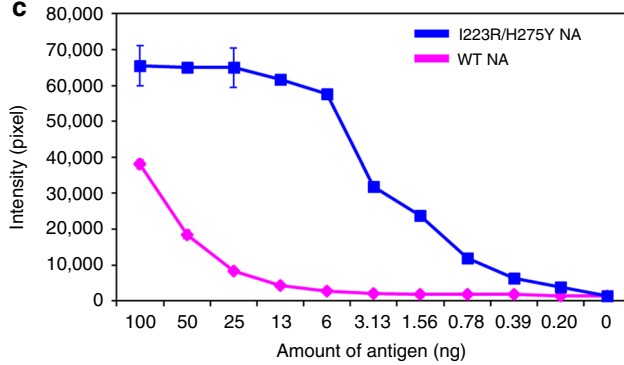

**Fig. 2 Analysis of the antigen-binding activity of A4. a** Affinity determination of A4 by SPR. Each experiment was repeated three times independently with similar results. **b** Specific interaction between A4 and NA by dot-blot analysis. A4 was applied to wt NA and I223R/H275Y NA dotted in a twofold dilution series (0.78–100 ng mL$^{-1}$), and HRP-conjugated anti-human IgG Fc was applied for detection. Each experiment was repeated three times independently with similar results. **c** The binding activity of A4 by densitometric analysis. All ELISA values were obtained from triplicate wells and are expressed as the mean. Error bars = standard deviation ($n = 3$).

Besides the increase in the number of hydrogen bonds, the strengthening of hydrophobic interactions may also be invoked to explain the higher binding affinity of the A4–I223R/H275Y NA complex than the A4–wt NA counterpart.

**Docking simulation results with mutational analysis.** In order to assess the importance of the hydrophobic interactions to stabilize the epitopes in the CDR of A4, we carried out the mutational analysis at positions His94 in the light chain and Trp33 in the heavy chain. These mutant A4 antibodies were purified with the same method as the wts. Supplementary Fig. 5 shows the binding affinities of I223R/H275Y NA with respect to the two

kinds of mutant A4 antibodies (H94A and W33A mutant A4 antibodies). We note that the mutation of A4 antibody at position 94 in the light chain from His to Ala leads to approximately 50-fold increase in the $K_d$ value associated with binding of I223R/H275Y NA (183 nM). This indicates the significant role of His94 in the light chain in the stabilization of the epitope. Similarly, the $K_d$ value of I223R/H275Y NA increases from 3.50 to 150 nM in going from the wt to the W33A mutant in the heavy chain. The results of mutational analyses are thus consistent with those of docking simulations indicating that the strengthening of hydrophobic interactions with aromatic side chains are responsible for tight binding of I223R/H275Y NA in the CDR of A4. Judging from the consistency between the experimental and computational results, the capability of forming a van der Waals contact with the aromatic residues in CDR seems to be a determinant for selective binding to A4 antibody.

**Application of A4 antibody to I223R/H275Y virus detection.** Development of bioreceptor is the key requirement of the biosensor production. Until now, antiviral drug-resistant influenza viruses were not routinely diagnosed because of the absence of the proper bioreceptor. We found that the A4 antibody exhibited highly increased binding capacity to the I223R/H275Y mutant virus. Since A4 is the first discovered antiviral multidrug-resistant influenza virus-specific antibody, it can be directly employed for the development of multidrug-resistant virus diagnostic methods. This prompted us to develop various kinds of I223R/H275Y mutant virus-sensing methods by using A4 antibody as shown in Fig. 4. First, A4 antibody was applied to the Au NPs to distinguish I223R/H275Y pH1N1 with wt pH1N1 in the naked eye. In the presence of I223R/H275Y pH1N1, color of A4-Au NPs changes from red to purple. Second, SERS-based immunoassay for I223R/H275Y mutant virus was demonstrated by using A4 antibody. SERS is a fascinating phenomenon that significantly increases the Raman signal of molecules in plasmonic hot spots. Since SERS has been employed for the sensitive detection of viruses due to its single-molecule sensitivity, the adoption of A4 in SERS-based immunoassay enables the sensitive detection of mutant virus. Third, A4-based LFA system was developed for the diagnosis of I223R/H275Y pH1N1. LFA is easy to use, cost-effective, and can diagnose influenza viruses rapidly, therefore A4-based LFA system can be used for the practical diagnosis of I223R/H275Y pH1N1 in real world.

**A4-based colorimetric detection of I223R/H275Y virus.** We designed an A4-based colorimetric assay to directly detect I223R/H275Y mutant virus in the naked eye. A4-Au NPs were generated by conjugating Au NPs to A4 via Au-S bond formation. The mean size of the A4-Au NPs was 28.5 ± 6.7 nm as determined by dynamic light scattering (DLS). The Au NPs showed maximum absorbance at 520 nm and were deep red in color. The A4-Au NPs remained monodispersed in the aqueous phase and deep red in color without aggregation (Supplementary Fig. 6A). As shown in Fig. 5a, the color of the A4-Au NP solution obviously changed from deep red to purple as the I223R/H275Y pH1N1 virus concentration increased because the A4-Au NPs aggregated owing to the binding interaction between A4 and the I223R/H275Y NA of the virus. Simultaneously, the ultraviolet (UV)–visible absorption spectrum was redshifted, with the absorbance at 520 nm (Abs$_{520}$) gradually decreasing and that at 580 nm (Abs$_{580}$) increasing (Supplementary Fig. 6B). The presence of wt pH1N1 virus and H275Y pH1N1 virus in the A4-Au NP reaction solutions caused little change in color or absorption spectral shift (Fig. 5a and Supplementary Figs. 6C and 7A). The absorbance ratio of these two wavelengths (Abs$_{580}$/Abs$_{520}$) was plotted versus various virus

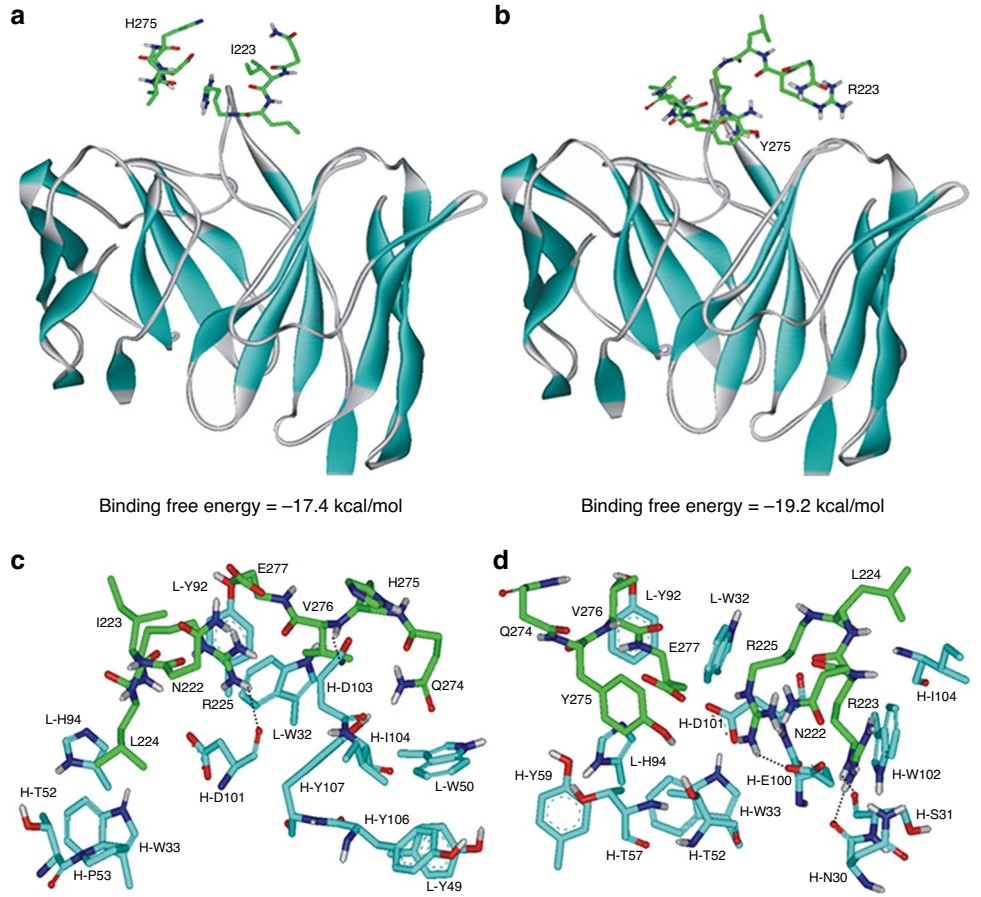

**Fig. 3 Structural feature of A4 and NA epitope.** Docked poses of the model epitopes for **a** wt NA and **b** I223R/H275Y NA in CDR of A4. Calculated binding modes of the model epitopes for **c** wt NA and **d** I223R/H275Y NA in CDR of A4. The β-strands of A4 and carbon atoms of NA epitopes are indicated in cyan and green, respectively. Dotted lines indicate the hydrogen bonds.

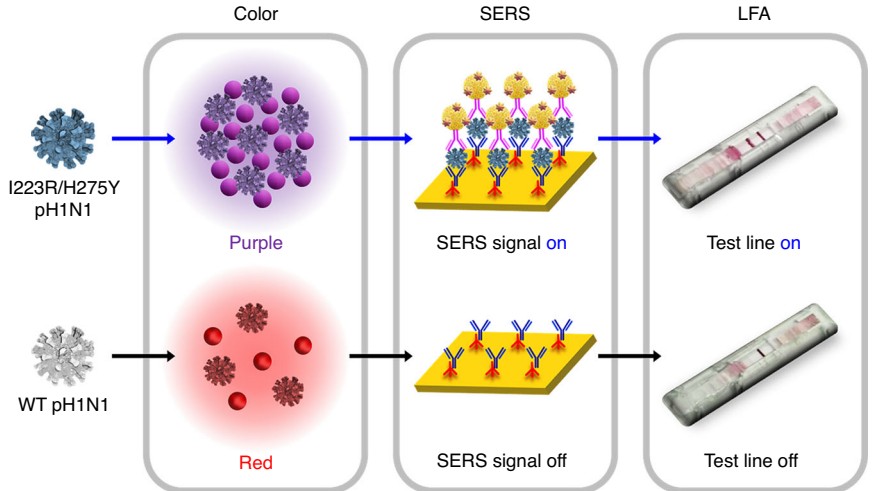

**Fig. 4 Application of A4 to I223R/H275Y virus detection.** Schematic illustration of I223R/H275Y pH1N1 detection using A4 in various sensing platforms including colorimetry, SERS, and LFA.

concentrations for quantitative analysis as shown in Fig. 5b. At maximum $Abs_{580}$ and $Abs_{520}$, the A4-Au NPs were purple-colored aggregates and red-colored dispersed particles, respectively. Thus, as the $Abs_{580}/Abs_{520}$ value increased, the solution became increasingly purple, indicating more aggregation. The $Abs_{580}/Abs_{520}$ values exhibited a linear correlation with the I223R/H275Y pH1N1 virus concentration. Even with $10^2$ PFU of

I223R/H275Y pH1N1 virus, the $Abs_{580}/Abs_{520}$ value was approximately 0.7. This indicates that the oseltamivir- and zanamivir-resistant I223R/H275Y pH1N1 virus can be detected directly by the naked eye using the A4-Au NPs.

**A4-based SERS detection of I223R/H275Y virus.** For the SERS-based immunoassay of I223R/H275Y pH1N1 virus, Au nanoplate

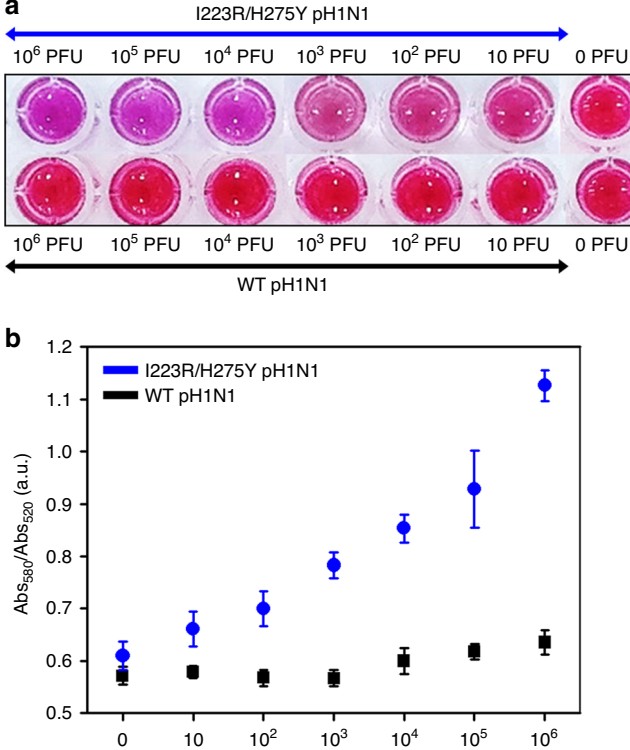

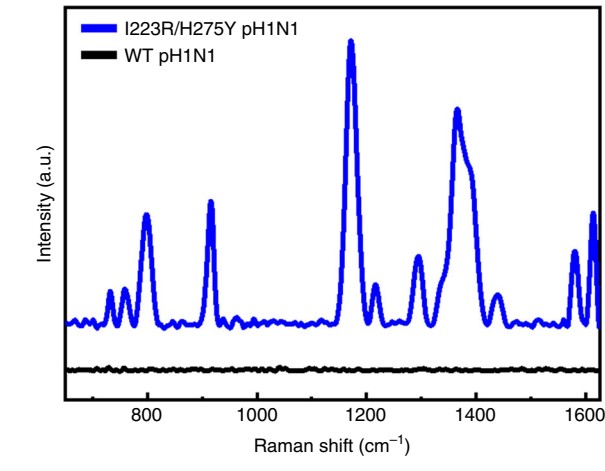

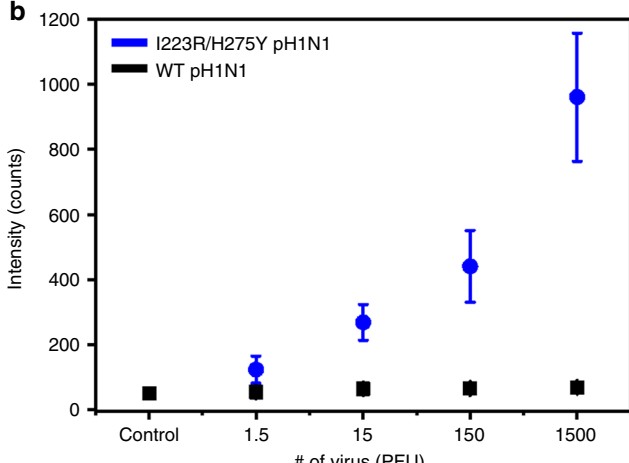

**Fig. 5 Colorimetric detection of I223R/H275Y pH1N1 using A4. a** Optical images of A4-Au NPs in the presence of I223R/H275Y pH1N1 (upper panel) and wt pH1N1 (lower panel). Each experiment was repeated three times independently with similar results. **b** Plot of $Abs_{580}/Abs_{520}$ versus the number of influenza virus. Data represent the average plus/minus standard deviation from three measurements.

**Fig. 6 SERS-based detection of I223R/H275Y pH1N1 using A4. a** SERS spectra of MGITC obtained from NPs-on-plate structures in the presence of I223R/H275Y pH1N1 and wt pH1N1. The number of both viruses is 1500 PFU. Each experiment was repeated ten times independently with similar results. **b** Plot of 1175 cm$^{-1}$ band intensity versus the number of influenza virus. Data represent the average plus/minus standard deviation from ten measurements.

and Au NP were selected as immune substrate and immunoprobe, respectively. The single-crystalline Au nanoplates can be excellent immune substrates due to their flatness and clean surfaces[31]. We prepared the Au nanoplates as reported previously[31] and immobilized three cysteine-tagged protein G (Cys3-protein G) and A4 antibodies onto the Au nanoplates sequentially. Protein G can bind to the Fc region of the antibodies and has been widely used to immobilize the antibodies with optimal orientation. The immunoprobes were prepared by modifying the Au NPs with malachite green isothiocyanate (MGITC) and hemagglutinin (HA) antibodies simultaneously. MGITC is a well-known Raman reporter and HA antibodies can bind to the both wt and mutant viruses. For the detection of influenza viruses, the immune substrates were reacted with I223R/H275Y pH1N1, wt pH1N1, or H275Y pH1N1, and then the immunoprobes were reacted (Supplementary Fig. 8). Figure 6a is the SERS-based immunoassay results for I223R/H275Y pH1N1 (blue spectrum) and wt pH1N1 (black spectrum). The number of both viruses is 1500 PFU. The SERS-based immunoassay result for H275Y pH1N1 is shown in Supplementary Fig. 7B. When the sample includes I223R/H275Y mutant virus, Au NPs on a nanoplate (NPs-on-plate) structures can be constructed through the immunoreaction of A4-I223R/H275Y pH1N1-HA. This NPs-on-plate architectures can provide significantly enhanced SERS signals. In contrast, very weak SERS signals were obtained when the sample has wt pH1N1 or H275Y pH1N1, because A4 does not bind to the wt influenza virus or single-mutant virus. This result was further confirmed by the scanning electron microscope (SEM) measurements of NPs-on-plate structures. Supplementary Fig. 9 shows the NPs-on-plate

structures, which we obtained from SERS spectra of Fig. 6a. The immunoprobes were well assembled on the Au nanoplate without aggregation after incubating in the mutant virus sample, whereas a few NPs were present on the nanoplate after incubating in the wt pH1N1 sample. Figure 6b shows the plot of 1175 cm$^{-1}$ band intensity versus the number of viruses. On increasing the number of mutant viruses, the SERS intensity increased. However, the intensities are negligible through the whole number of wt viruses. The corresponding SEM images also agree well with the SERS results (Supplementary Fig. 10). The numbers of immunoprobes increased by increasing the concentration of mutant virus but a few immunoprobes were observed regardless of the concentration of wt pH1N1. By combining the developed A4 antibody and SERS-based immunoassay, we could detect I223R/H275Y mutant virus as low as 1.5 PFU.

**A4-based LFA of I223R/H275Y virus.** LFA, a most widely used point-of-care testing format, allows the rapid, visual, and direct analysis of targets with high specificity and sensitivity. Currently available LFAs for influenza viruses do not provide information on the antiviral drug resistance of viruses. We produced an LFA

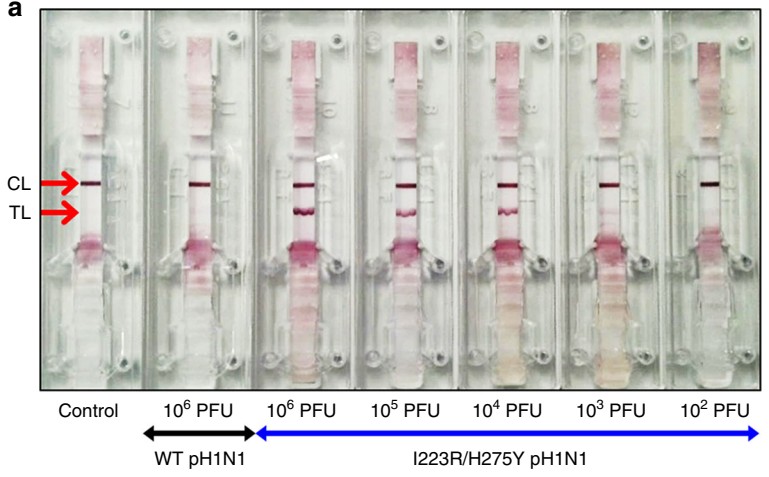

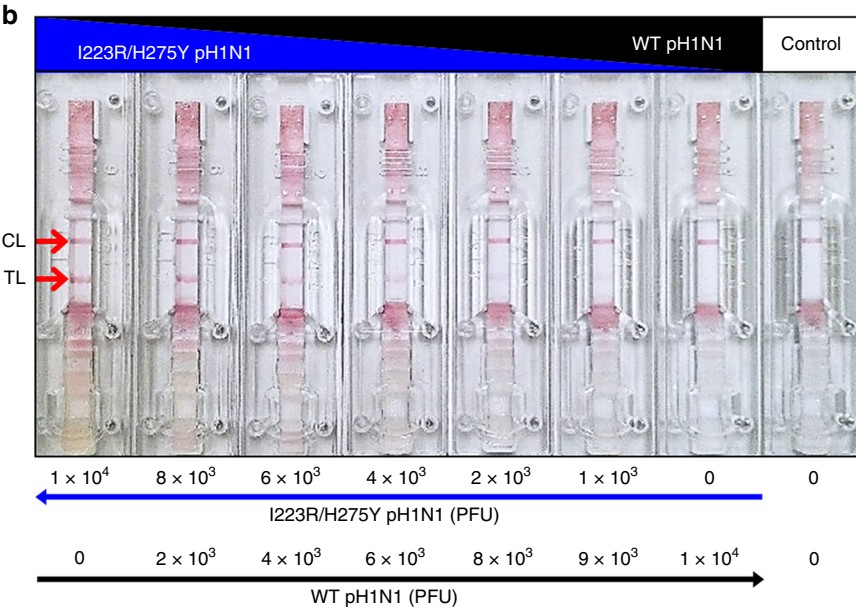

**Fig. 7 LFA of I223R/H275Y pH1N1 using A4. a** Optical images of A4-based lateral flow systems after detection of I223R/H275Y pH1N1, wt pH1N1, and control sample. **b** Optical images of A4-based lateral flow systems after detection of I223R/H275Y and wt pH1N1 mixture in nasal fluid and control sample.

for multidrug-resistant influenza virus by employing A4 antibody. Briefly, A4-Au NPs were added to the conjugation pad, and the capture antibody and goat anti-human IgG were immobilized on the NC membrane as the test and control lines, respectively. Therefore, two lines in the test indicate the presence of I223R/H275Y pH1N1 virus. After the assembly of LFA components, virus sample was applied to the sample pad. The antiviral-resistant I223R/H275Y pH1N1 virus reacts and binds to A4-Au NPs, and then the complex is captured by immobilized anti-influenza A nucleoprotein IgG on the test line, resulting in a positive signal. On the other hand, wt pH1N1 virus and H275Y pH1N1 virus are not able to interact with A4-Au NPs; thus a signal from the test line is not observed. Figure 7a and Supplementary Fig. 7C are the micrographs of the LFAs after detection of wt, double-mutant, and single-mutant viruses. When the I223R/H275Y pH1N1 virus samples were applied, the red test lines were observed clearly. Importantly, even in the cases of high concentrations of wt pH1N1 virus ($10^6$ PFU) and H275Y virus ($10^6$ PFU), only the control line was observed in the absence of the I223R/H275Y pH1N1 virus. The sensitivity of A4-based LFA for the detection of I223R/H275Y pH1N1 virus was tested at various virus concentrations ($10^6$–$10^2$ PFU), and positive signals

were confirmed for samples containing $10^6$, $10^5$, $10^4$, and $10^3$ PFU of I223R/H275Y pH1N1 virus in a concentration-dependent manner. In addition, the dose effect of I223R/H275Y detection in nasopharyngeal swab samples was demonstrated (Supplementary Fig. 11). These results demonstrate that this A4-based LFA has a high detection specificity depending on the dose of I223R/H275Y pH1N1 virus, with an limit of detection (LOD) of $10^3$ PFU per test.

The diagnostic efficiency and specificity of the A4-based LFA was further verified by treating a mixture of antiviral-susceptible wt pH1N1 and oseltamivir- and zanamivir-resistant I223R/H275Y pH1N1 viruses at various ratios under human nasal fluid conditions (Fig. 7b). The total virus concentration was $1 \times 10^4$ PFU per test ( = $1 \times 10^5$ PFU mL$^{-1}$), and virus-free buffer was used as the control. As the concentration of I223R/H275Y pH1N1 virus decreased, the detection signal in the test line gradually decreased from strong to weak. The signal from the test line was observed even in a mixed sample containing a high concentration of the wt pH1N1 virus ($9 \times 10^3$ PFU) and a small concentration of the I223R/H275Y pH1N1 virus ($1 \times 10^3$ PFU). No signal was observed in the test line for the sample containing only the wt pH1N1 virus ($1 \times 10^4$ PFU) and not the I223R/H275Y pH1N1

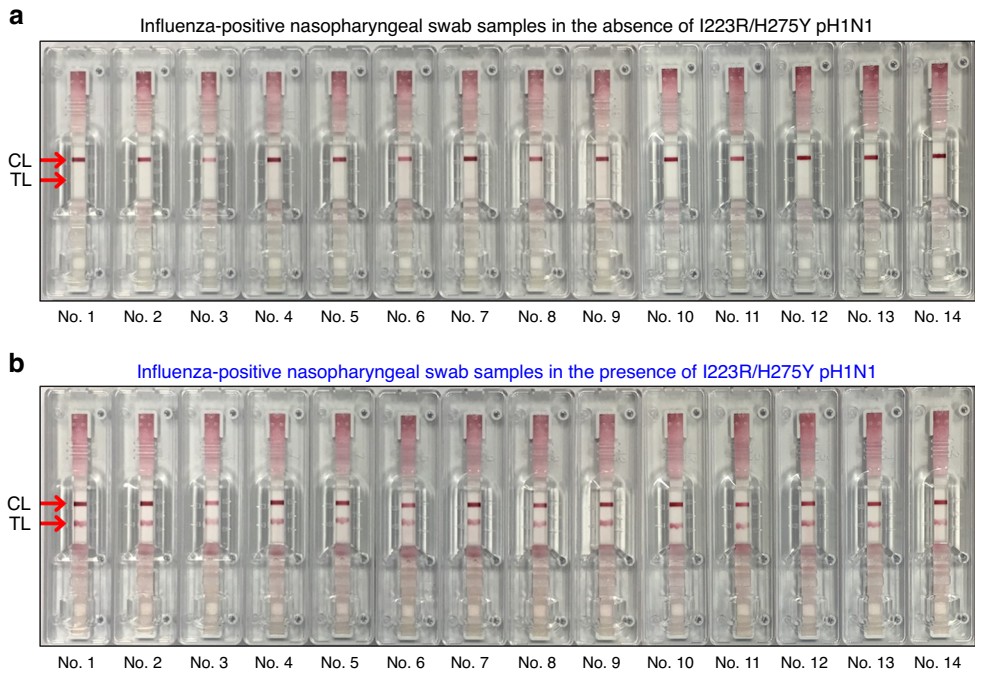

**Fig. 8 LFA of I223R/H275Y pH1N1 in human nasal fluid samples using A4.** Optical images of A4-based lateral flow systems after detection of influenza-positive nasopharyngeal swab samples **a** in the absence or **b** presence of I223R/H275Y pH1N1 ($10^3$ PFU).

virus. These results demonstrate that A4-based LFA has a high detection specificity depending on the dose of mutant I223R/H275Y pH1N1 virus even in the mixture with wt virus, with an LOD of $10^3$ PFU per test. Considering that the RIDTs are mainly performed by using the human nasal samples including small amounts of mutant viruses and overwhelming amounts of wt viruses, this result is quite impressive.

Finally, we investigated that the A4-based LFA can be used for the diagnosis of multidrug-resistant influenza virus in real human samples. Because the current diagnosis in the hospital does not confirm the presence of antiviral drug resistance of influenza viruses, nasopharyngeal samples from patients with influenza-like symptoms were collected and mixed with I223R/H275Y pH1N1 virus ($10^3$ PFU). Figure 8a is an optical image of A4-based lateral flow systems after detection of influenza-positive nasopharyngeal swab samples in the absence of I223R/H275Y pH1N1 virus where only the control line was observed. On the other hand, the red test lines were observed clearly after detection of influenza-positive nasopharyngeal swab samples in the presence of I223R/H275Y pH1N1 virus (Fig. 8b). Totally, we tested 40 human nasopharyngeal swab samples (Supplementary Fig. 12). The sensitivity and specificity of A4-based LFA developed for rapid antiviral multidrug-resistant influenza virus diagnostic tests are 100% (40/40) and 100% (40/40), respectively. According to the guideline of National Institute of Food and Drug Safety Evaluation of Korea, a minimum of 20 sample results are required for approval of an in vitro diagnostic system. We also tested the present mutant virus-sensing methods by using four kinds of influenza virus subtypes (A/Puerto Rico/8/1934 (H1N1), A/Brisbane/10/2007 (H3N2), A/swine/Korea/GC0503/2005 (H1N1), and A/canine/Korea/MV1/2012 (H3N2)). The virus samples were prepared by mixing each of the I223R/H275Y influenza viruses ($10^3$ PFU) with influenza-positive nasopharyngeal swab samples ($n = 6$). Figure 9 and Supplementary Fig. 13 are optical images of A4-based lateral flow system after detection of influenza-positive nasopharyngeal swab samples in the absence or presence of I223R/H275Y influenza viruses, in which positive test lines were clearly observed in influenza-positive

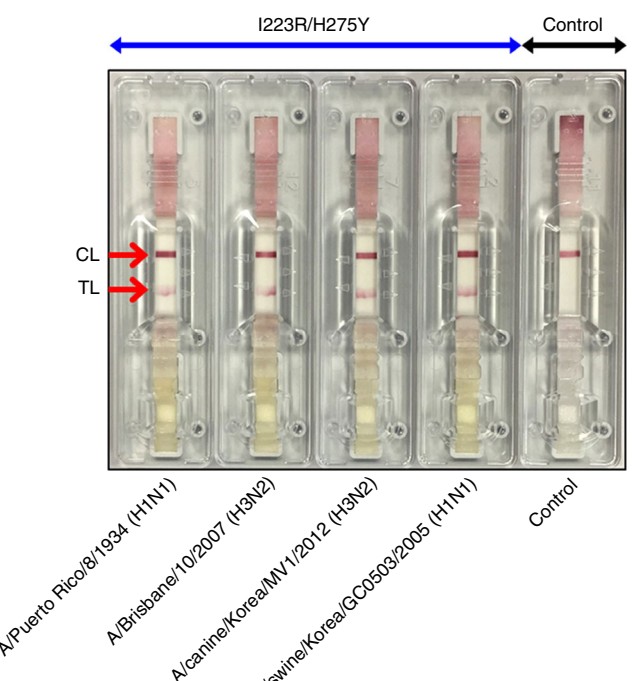

**Fig. 9 LFA of I223R/H275Y influenza viruses in human nasal fluid samples using A4.** Optical images of A4-based lateral flow systems after detection of influenza-positive nasopharyngeal swab samples in the absence or presence of I223R/H275Y influenza viruses (A/Puerto Rico/8/1934 (H1N1), A/Brisbane/10/2007 (H3N2), A/swine/Korea/GC0503/2005 (H1N1), and A/canine/Korea/MV1/2012 (H3N2)).

nasopharyngeal swab samples taken with various subtypes of I223R/H275Y influenza virus. Based on this result, we anticipate that the A4-based LFA can be used for the rapid antiviral multidrug-resistant influenza virus diagnostic test in near future.

The accurate diagnosis of an infectious disease has a significant impact on clinical decisions made in response to the disease. Drug susceptibility testing prior to therapy will prevent drug abuse and inhibit the spread of drug-resistant viruses. In this study, we developed an A4 antibody specifically recognizing and binding to the mutant I223R/H275Y NA and demonstrated that it is possible to directly detect antiviral multidrug-resistant viruses in various sensing platforms. We have verified the applicability of A4 antibody for the development of quantitative detection tools on a variety of detection platforms, including naked-eye detection, SERS-based immunoassay, and lateral flow system. The development of the A4 antibody in this study enables fast, simple, and reliable point-of-care assays of antiviral multidrug-resistant influenza viruses. In addition to current influenza virus infection testing methods, diagnostic tests for antiviral multidrug-resistant viruses will improve clinical judgment in the treatment of influenza virus infections, avoid the unnecessary prescription of ineffective drugs, and greatly improve current therapies.

## Methods

**Materials**. All chemicals were purchased from Sigma-Aldrich unless otherwise specified. NA-Fluor™ Influenza NA Assay Kit was purchased from Applied Biosystems. Viral RNA Extraction Kit was purchased from Qiagen. Absorbent pads (CFSP203000, cellulose fiber) and NC membrane (HF090MC100) were purchased from Millipore. A polyester conjugate pad was purchased from Boreda Biotech. Anti-influenza A virus nucleoprotein antibody were purchased from Abcam. Influenza virus strains, including A/California/07/2009 (H1N1), A/Puerto Rico/8/1934 (H1N1), and A/Brisbane/10/2007 (H3N2), were originally provided from the Korea Centers for Disease Control and Prevention, Korea. A/swine/Korea/GC0503/2005 (H1N1) and A/canine/Korea/MV1/2012 (H3N2) were kindly provided by Professor Dae-Sub Song (Korea University). The pH1N1/H275Y mutant virus (H275Y mutation A/Korea2785/2009 pdm: NCCP 42017) was obtained from the National Culture Collection for Pathogens (NCCP) operated by the Korea National Institute of Health. The titers of viruses were measured by an RT-PCR assay from Promega according to the manufacturer's instructions.

**Virus propagation and clinical nasopharyngeal samples**. Madin-Darby Canine Kidney (MDCK) cells (ATCC CCL-34) were obtained from the American Type Culture Collection (ATCC) (Rockville, USA). MDCK cells were cultivated in Dulbecco's Modified Eagle's Media supplemented with 5% fetal calf serum. The cells were infected with influenza virus strains, and the virus titer (PFU) was determined in the presence of overlay media containing agarose and tosyl phenylalanyl chloromethyl ketone–trypsin for 72 h at 37 °C and 5% CO$_2$. The reverse genetics system was used to generate different subtypes of the I223R/H275Y influenza virus. The expression plasmids for the eight-plasmid reverse genetic system were kindly donated by Dr. Meehyein Kim of Korea Research Institute of Chemical Technology (Daejeon, Republic of Korea)[32,33]. NA and HA genes derived from A/Puerto Rico/8/1934 (H1N1), A/Brisbane/10/2007 (H3N2), A/swine/Korea/GC0503/2005 (H1N1), and A/canine/Korea/MV1/2012 (H3N2) genomic RNAs were cloned individually into pVP-NA or pVP-HA vector using universal reverse primers and genome-specific primers. In addition, I223R/H275Y mutation was introduced into the NA fragment by the forward primer: 5′-TATTCGTCT CAGGGATGAAGACTATCATTGCTTTGAGCTACATT-3′ and reverse primer: 5′-ATATCGTCTCGTATTTGTTTTTAATTAATGCACTCAAATGCAAA-3′. Briefly, co-cultured 293T and MDCK cells (0.5 × 10$^6$ cells per well) grown in 6-well plate were transfected using Lipofectamine 3000 (Thermo Fisher Scientific) according to the manufacturer's instructions (1 µg of each influenza DNA plasmid). After 6 h of incubation, the transfection medium was replaced by Opti-MEM. After 30 h of transfection, each well was supplemented with 1 ml of Opti-MEM containing 0.75 µg mL$^{-1}$ tosylsulfonyl phenylalanyl chloromethyl ketone–trypsin. At 3–6 days post transfection, cell supernatants were titrated onto MDCK cell monolayers to estimate influenza virus titers. All experiments were done in triplicate.

Nasopharyngeal samples from patients with influenza-like symptoms were collected with the flocked nasopharyngeal swabs and placed into the virus transport media (UTM, Copan Diagnostics Inc., USA). Each sample was analyzed with the AdvanSure RV RT-PCR Kit (LG Life Sciences, Korea) following the manufacturer's instructions with the SLAN Real-Time Quantitative PCR Detection System (LG Life Sciences, Korea). A Ct (threshold cycle) value of 25 was used for the cut-off for influenza positivity according to the manufacturer's recommendation. All samples were stored at −70 °C before use. The protocol for this retrospective study was reviewed and approved by the Institutional Review Board of Yonsei University Health Service, Severance Hospital, Seoul, Korea (IRB approval number: 4-2017-1179).

**Expression and purification of recombinant protein**. The wt NA and mutant NA (I223R/H275Y) genes derived from A/H1N1 2009 pandemic influenza virus were amplified by PCR. The PCR products were cloned into the pAcGP67A vector (BD Biosciences) and transfected into the Sf9 insect cell line to produce recombinant proteins. The culture medium was collected, clarified by centrifugation at 110 × g, and subjected to further viral propagation. The cultured Sf9 cells were inoculated with recombinant baculovirus at a multiplicity of infection of 10 and incubated at 27 °C for 96 h. The cells were collected after centrifugation, and the cell pellet was resuspended in cell lysis buffer (50 mM Tris-HCl, pH 8.5, 5 mM 2-mercaptoethanol, 100 mM KCl, 1 mM phenylmethylsulfonyl fluoride), sonicated, and centrifuged at 16,000 × g for 15 min at 4 °C. After the cell lysate was sonicated to reduce its viscosity, the cell debris was removed by centrifugation for 1 h at 16,000 × g. The soluble protein from the cell supernatant was applied to Ni-nitrilotriacetic acid agarose resin (Qiagen), washed, and eluted with buffer (50 mM Tris-HCl, 0.5 M NaCl, 0.5 M imidazole, pH 8.0). The purified proteins were dialyzed against phosphate-buffered saline (PBS) and further purified by Q-Sepharose anion-exchange chromatography (GE Healthcare).

**A4 panning and screening**. For antibody screening, large naive human antigen-binding fragment phage display library (3 × 10$^{10}$) in Korea Research Institute of Bioscience and Biotechnology was used[34]. A phagemid vector (KRIBB-Fab) contains a bicistronic operon under the control of *LacZ* promoter. Immunotubes (Nunc) were coated with 100 µg of wt NA or I223R/H275Y NA overnight at 4 °C, washed twice with PBS, and blocked with 4% skim milk in PBS at 37 °C for 1 h. The antibody library phages were preincubated with wt NA at 37 °C for 2 h. The subtracted phages were then incubated with I223R/H275Y NA at 37 °C for 1 h. After washing with 0.05% Tween 20 in PBS (PBST), bound phages were eluted with 0.1 M glycine–HCl (pH 2.2) and neutralized with 2 M Tris base. The eluted phages were amplified by infecting TG1 cells followed by superinfection with helper phages (VCSM13)[35]. The amplified phages were then subjected to another round of panning. Four rounds of panning were conducted, and the stringency of selection was increased with each round by gradually increasing the number of washes from 10 to 40.

To screen individual clones for specific binding to I223R/H275Y NA, 500 colonies were randomly selected from the output plate after the third or fourth round of panning, cultured in Superbroth medium containing 100 µg mL$^{-1}$ ampicillin until optical density of 0.5, and induced for Fab expression in *Escherichia coli* TG1 cells at 30 °C overnight by adding isopropyl β-D-1-thiogalactopyranoside to a final concentration of 1 mM. The culture supernatant of each clone was subjected to ELISA to screen anti-I223R/H275Y NA antibodies. In detail, a microtiter plate was coated with 100 ng of I223R/H275Y NA in coating buffer (0.05 M carbonate buffer, pH 9.6) and incubated at 4 °C overnight. After blocking, Goat F(ab′)2 Anti-Human IgG (Fab′)2-HRP (Abcam, 1:1000) antibody was used for the colorimetric detection of bound clones using the tetramethylbenzidine substrate. Clones showing positive signals in ELISA were subjected to DNA sequencing, and the nucleotide sequences of variable heavy chain (VH) and variable kappa light chain (VK) regions were determined.

**Expression and purification of whole IgG**. To convert the selected Fabs into whole IgG format, the VH and VK sequences were amplified by PCR and combined with the leader sequences of IgG heavy and light chains, respectively, by overlap extension PCR using Pfu DNA Polymerase (Thermo Scientific). The VH and VK with leader sequences were sequentially cloned into the *Eco*RI-*Apa*I and *Hin*dIII-*Bsi*WI sites, respectively, in the antibody expression cassette (pdCMV-dhfrC) containing the human constant region of γ1 heavy chain (Cγ1) and Cκ gene[36]. For transient IgG expression, the resulting expression plasmid was introduced into HEK293T cells using Lipofectamine (Invitrogen) according to the manufacturer's instructions, and the transfected cells were cultured in protein-free medium (CD293, Invitrogen). The IgG was purified from the culture supernatant by affinity chromatography on protein A (Millipore), and its expression and purity were analyzed by western blotting and SDS-PAGE, respectively. The protein concentration was determined by UV spectrophotometry (NanoDrop; Thermo Fisher Scientific).

**Enzyme-linked immunosorbent assay**. Microtiter wells were coated with the purified NA (100 ng) in 50 mM sodium carbonate buffer (pH 9.6) at 4 °C overnight, blocked with bovine serum albumin (BSA) (2%) in PBS, and washed with PBST. A reaction mixture containing purified antibody (10 nM) and various concentrations (10$^{-11}$–10$^{-5}$ M) of NA as a competing antigen were preincubated at 37 °C for 1–2 h. The mixture was then added to each well previously coated with 100 ng of NA. Horseradish peroxidase (HRP)-conjugated goat anti-human IgG (Pierce, 1:3000) was used for the detection of bound IgG. Color was developed with the 3,3′,5,5′-tetramethylbenzidine substrate reagent set (BD Biosciences), and the absorbance at 450 nm was measured using a microtiter plate reader (Emax; Molecular Devices). Affinity was determined as the antigen concentration required to inhibit 50% of binding activity and binding affinity ($K_d$) value was calculated from a Klotz plot.

**SPR analysis**. SPR experiments were performed to determine the binding affinity between the target proteins and antibodies on a BIAcore 3000 (GE Healthcare). A4 antibody (7 μg mL$^{-1}$) was covalently immobilized on sensor chip CM5 by standard amine coupling protocol. Briefly, a mixture of 1-ethyl-3-(3-dimethylaminopropyl)-carbodiimide (0.2 M) and N-hydroxysuccinimide (0.05 M) was injected onto the chip at a flow rate of 5 μL min$^{-1}$, and various concentrations of proteins (0–2000 nM) were injected across the sensor surface at flow rate 5 μL min$^{-1}$. Data were fitted to a 1:1 Langmuir-binding model using the BIAevaluation software provided by the manufacturer for the kinetic analysis.

**Homology modeling of A4 antibody**. 3D structure of A4 antibody was obtained through the homology modeling with MODELLER program[37] using the X-ray crystal structure of the G6 antibody fragment specific for binding to VEGF[38]. G6 antibody served as a template for building the structure of A4 antibody suitable for docking simulations with epitopes (Supplementary Fig. 4). During the homology modeling, we adopted the conjugate gradient methods for structural optimizations and the molecular dynamics simulations to minimize the violations of spatial restraints. The atomic coordinates for the gap regions were optimized from a randomized distorted structure to connect the two anchoring positions, as implemented in MODELLER program. The loop structures were modeled with the enumeration algorithm to enhance the accuracy of the predicted structure[39].

**Docking simulations of wt and I223R/H275Y NA in the CDR of A4**. 3D structure of A4 obtained in the precedent homology modeling served as the receptor model in docking simulations with wt and I223R/H275Y mutant NA. The epitope structures were extracted from the X-ray crystal structure of (PDB entry: 4B7R)[40]. Docking simulations were carried out using the AutoDock program to estimate the binding free energy ($\Delta G_{bind}$) of the epitope in the CDR of A4, which can be expressed mathematically as follows[41].

$$
\begin{aligned}
\Delta G_{bind} = & W_{vdW} \sum_i \sum_{j>i} \left( \frac{A_{ij}}{r_{ij}^{12}} - \frac{B_{ij}}{r_{ij}^{6}} \right) + W_{hbond} \sum_i \sum_{j>i} E(t) \left( \frac{C_{ij}}{r_{ij}^{12}} - \frac{D_{ij}}{r_{ij}^{10}} \right) \\
& + W_{elec} \sum_i \sum_{j>i} \frac{q_i q_j}{\varepsilon(r_{ij}) r_{ij}} + W_{tor} N_{tor} + W_{sol} \sum_i^{atoms} S_i \left( O_i^{max} - \sum_{j \neq i}^{atoms} V_j e^{-\frac{r_{ij}^2}{2\sigma^2}} \right).
\end{aligned}
\tag{1}
$$

The weighting parameters for van der Waals contacts ($W_{vdW}$), hydrogen bonds ($W_{hbond}$), electrostatic interactions ($W_{elec}$), entropic penalty ($W_{tor}$), and ligand dehydration free energy ($W_{sol}$) were set to 0.1485, 0.0656, 0.1146, 0.3113, and 0.1711, respectively, as in the original AutoDock program. $r_{ij}$ stands for the interatomic distance, and $A_{ij}$, $B_{ij}$, $C_{ij}$, and $D_{ij}$ are associated with the well depth and the equilibrium distance in the potential energy function. The hydrogen bond term has the additional weighting factor ($E(t)$) to describe the angle-dependent directionality. To compute the electrostatic interaction energy between A4 antibody and the epitopes, we used the sigmoidal function with respect to $r_{ij}$ proposed by Mehler et al. as the distance-dependent dielectric constant[42]. In the entropic penalty term, $N_{tor}$ indicates the number of rotatable bonds in the epitope. In the hydration free energy term, $S_i$ and $V_i$ denote the atomic solvation energy per unit volume and the fragmental atomic volume, respectively, while Occ$_i^{max}$ represents the maximum occupancy of each atom in the epitope[43]. All the energy parameters in Eq. (1) were extracted from the original AutoDock program to derive the binding modes of wt and I223R/H275Y mutant NA in the CDR of A4.

Among 20 conformations generated with the genetic algorithm, those clustered together had similar binding modes differing by <1.5 Å in positional root-mean-square deviation. The lowest-energy configuration in the top-ranked cluster was selected as the final structural models for antigen–antibody complexes.

**Colorimetric detection of antiviral multidrug-resistant influenza virus**. To prepare A4-conjugated Au NPs, the A4 antibody (10 μg) was added to a mixture of 1 mL of a colloid of Au NPs and 0.1 mL of borate buffer (0.1 M, pH 8.5). After incubation for 30 min at room temperature, 0.1 mL of 1% BSA was added to block the surface of the Au NPs. After incubation for 60 min at 4 °C, the conjugates were collected by centrifugation at 16,000 × g for 15 min at 4 °C. The final conjugates were resuspended in deionized waster. The changes in size of the particles were confirmed by using DLS (ELS-Z, Otsuka Electronics). The color change was observed and measured using a multidetection microplate reader (Cytation 5, BioTek).

**SERS-based detection of antiviral multidrug-resistant influenza virus**. Au nanoplates were synthesized in a horizontal hot-wall single-zone furnace system with a 1-inch diameter inner quartz tube. The system was equipped with pressure and mass flow controllers. In a quartz tube, an Au slug-containing alumina boat was placed at the center of a heating zone. Before the reaction, the quartz tube was purged with N$_2$ gas for 30 min to maintain an inert atmosphere, and the pressure was lowered to 5−10 Torr with a gas flow rate of 100 sccm. The Au slug-containing alumina boat was heated to 1100−1170 °C, and the reaction time was 60−90 min. During the reaction, Au vapor was transported by the carrier gas from the high-temperature zone to the low-temperature zone, where substrates were present. For the preparation of immune substrate, the Au nanoplates on silicon wafer were

incubated with 5 nM Cys3-protein G solution in PBS buffer for 12 h at 4 °C and rinsed three times with 0.1× PBS. Cys3-protein G-modified Au nanoplates were immersed in the 1 nM A4 antibody solution in PBS buffer for 12 h at 4 °C and rinsed three times with 0.1× PBS. For the preparation of immunoprobes, 100 μL of MGITC (1 μM in ethanol) solution was added to the Au NP solution to prepare a 1-mL solution. Next, the Au NP solution was incubated for 45 min at room temperature with orbital shaking. Simultaneously, we mixed 10 μL of HS-(CH$_2$)$_{10}$-NHS solution (10 μM in tetrahydrofuran) and 100 μL of HA antibody solution (100 ng mL$^{-1}$ in PBS) for 45 min at room temperature with orbital shaking. Fifty microliters of the mixed solution was added to the prepared Au NP solution, and this solution was incubated for 45 min at room temperature with orbital shaking. After that, 1.15 μL of 1 M Tris-HCl buffer (pH 6.8) was added to the Au NP solution for 30 min at room temperature with orbital shaking. Finally, the Au NP solution was centrifuged (19,800 × g) for 15 min and re-suspended in 0.1× PBS.

The I223R/H275 and wt viruses were diluted in PBS buffer with various concentration and added to the prepared immune substrates for 6 h at 4 °C with shaking. Then the immune substrates were rinsed three times with 0.1× PBS and incubated in the prepared Au NP solution for 45 min at room temperature. After washing with distilled water and drying with N$_2$ gas, SERS spectra and SEM images were obtained from the samples. SERS measurements were carried out using a LabRAM HR system (HORIBA Jobin Yvon, France). The excitation source was a He–Ne laser operating at $\lambda = 633$ nm, and the laser spot was focused on a sample through a 50× objective lens. The SERS signals were recorded with a thermo-dynamically cooled electron multiplying charge-coupled device (Andor) mounted on a spectrometer with a 1200-groove mm$^{-1}$ grating. SEM images were obtained using a Nova230 system at an accelerating voltage of 15 keV.

**LFA of antiviral multidrug-resistant influenza virus**. The LFA test strip consists of four components: a sample pad, an NC membrane, a conjugation pad, and an absorbance pad. The sample pad was saturated with PBS solution containing 0.2% Tween-20 and 1% (w/v) BSA. The sample pad was dried at room temperature and stored in a desiccator for future use. The colloidal Au NP probes were added to the conjugation pad, dried at room temperature for 2 h, and then stored in a desiccator. The anti-influenza A nucleoprotein IgG (2 mg mL$^{-1}$) and goat anti-human IgG (1 mg mL$^{-1}$) were immobilized on the NC membrane as the test and control lines, respectively. After the NC membranes were dry, the membranes were blocked with 1% BSA. After all the components were assembled on the backing card, the test strip was assembled on a strip cassette. During the assays, 150 μL of the sample was directly applied to the sample pad, and this volume spread across the whole strip within 5 min. The LFA was also evaluated in virus-containing 10% nasal fluid (Lee Biosolution, Inc.) as well as in virus mixtures prepared by mixing double mutant and wt viruses in a same way as described above. Lastly, the LFA was examined in the influenza-positive nasopharyngeal swab samples.

**Reporting summary**. Further information on research design is available in the Nature Research Reporting Summary linked to this Article.

## Data availability

All data generated or analyzed during this study are included in this published Article and its Supplementary Information files, and the source data underlying all figures and Supplementary Figures are provided as a Source Data file.

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

## Acknowledgements

This research was supported by the Center for BioNano Health-Guard funded by the Ministry of Science and ICT (MSIT) of Korea as Global Frontier Project (HGUARD_2013M3A6B2078950 and HGUARD_2014M3A6B2060507), the Bio & Medical Technology Development Program of the National Research Foundation (NRF) funded by MSIT of Korea (NRF-2018M3A9E2022821), the Basic Science Research Program of the NRF funded by MSIT of Korea (NRF-2018R1C1B6005424, NRF-2020R1A2C1010453 and NRF2019R1C1C1006867), and KRIBB Initiative Research Program.

## Author contributions

K.G., Hyeran Kim, Y.-A.C., S.G.H., G.H., H.-N.K., and J.J. contributed to the development of A4 antibody; K.G., Hyeran Kim, S.G.H., G.H., H.-N.K., E.-K.L., and J.J. contributed to the colorimetric and LFA experiments; M.L., Hongki Kim, and T.K. contributed to the SERS experiments; H.P. contributed to the structural characterization of A4 antibody; D.Y. supported the nasopharyngeal samples from patients with influenza-like symptoms; T.K. and J.J. contributed to the manuscript writing.

## Competing interests

The authors declare no competing interests.
