## [Peer Review File · Nature Communications]

Reviewers' Comments:

Reviewer #1:

Remarks to the Author:

In this work, the authors develop a new monoclonal antibody capable of specifically recognizing an influenza neuraminidase double-mutant which confers drug resistance. The goal stated by them is to create a point-of-care fast diagnosis method for drug-resistant strains. This is a real problem identified by the authors and they go in the direction of solving it.

This is a very respectable work but, despite the paper's merits, it cannot be framed as a protein engineering paper neither as diagnostic development paper since it lacks in both areas. I find that there are major issues that have to be addressed to substantiate their claims before it can be accepted in a publication such as Nature Communications.

1. They show the antibody recognizing the double-mutant protein and not recognizing the wild-type. However, they never use single-mutants in their study which also confers drug resistance and is more prevalent than the double mutants. It's paramount to test the single mutants to show the diagnostic value of the antibody.

2. They frame the paper as a diagnostic development work, but all the tests are done against recombinant protein or laboratory produced viruses. It's paramount that their very interesting tools are tested on real human samples where it should be compared to traditional diagnostic methods such as DNA sequencing and/or qPCR to prove its ability to accurately differentiate the virus in real samples. Sensitivity and specificity numbers should be calculated after these tests to show the validity of the methods.

3. They show computational studies for docking of the antibody-antigen complex without providing any methodological detail. Free energy number is discussed but it's never explained how the numbers were obtained. Different from protein structure modeling and despite its evolution in the past years, protein-protein docking methods are still only valid after experimental validation. Given two structures, a docking algorithm will always find many docking conformations even if the two proteins don't actually bind. Without experimental validation of the docking structure, it's impossible to assess its quality and validity. Without further evidence, there is no way to substantiate the claims made. Moreover, all the features pointed out for increased binding affinity consider hypothetical enthalpic changes, when in fact it could be entropic contributions.

4. There are no considerations or references on phage display library specifications. This is very important for an antibody display paper.

Those are the major points that I firmly believe should be addressed to make this paper, that has a lot of potentials, a very important piece of work that will push the current state-of-the-art on influenza diagnosis. Other minor points are described below

- Affinities measured by ELISA are used for a rough estimation. Since the authors measured the affinity of the antibody by surface plasmon resonance, which is the state-of-the-art technique for such, the ELISA measurements should be kept as supplemental information or omitted since it does not add anything to the paper. Moreover, the methodology for measuring the affinity by ELISA was not described in the material and methods section.

- There is no methodology for docking described in the paper and also there is no methodology for free energy calculation described in the paper. Please clarify this.

- Different from protein structure modelling and despite its evolution in the past years, protein-protein docking methods are still only valid after experimental validation. Given two structures, a docking algorithm will always find many docking conformations even if the two proteins don't actually bind. Without experimental validation of the docking structure, it's impossible to assess its quality and validity. Without further evidence there is no way to substantiate the claims made. Moreover, all the features pointed out for increased binding affinity consider hypothetical enthalpic changes, when in fact it could be entropic contributions.

Line 34: It abbreviates neuraminidase to "NA" which has never been used in the text and so far.

Line 121: please correct Delbecco to Dulbecco

Line 130: why did you used insect expression system. Instead of mammalian expression system?

Line 138: the centrifugation for 1h at 16,000g was to precipitate the protein or cell debris? I assume the protein was in the supernatant and that was used for affinity chromatography purification, but the text suggests that the "protein was protein was obtained by centrifugation" which implies that the protein was pelleted.

Line 146: where the antibody library comes from? Is there a reference for this library? Is it human? Is it scFv or Fab format? What is the promoter? What is the helper phage used?

Line 147: How many washes were performed in each round? It's very important for the authors to give details of the method used to obtain the antibody so others can successfully follow.

Line 156: How was the soluble Fab expressed? Was it fused to g3p? Is there an amber stop codon between the Fab gene and g3p so one can express soluble Fab when using a non-suppressor strain?

Line 157: how was the Fab detected? Is there a tag so one can use a labeled anti-tag antibody? Please give more details.

Line 160: Did the library only contain kappa light chains and no lambda light chains?

Line 164: What polymerase was used? Please insert in the text.

Line 165: Is recombinant PCR the same as overlap extension PCR? If so, I believe the later term is more widely used and well understood.

Line 166: The fragments were not subcloned, but merely cloned. Subcloning is the technique where a fragment is directly excised with restriction enzyme from one plasmid and the ligated into another. In this case, the fragments were amplified by PCR, modified and then cloned.

Line 176: What was the Bioanalyzer used for? It's not clear. It looks like it was used for western blotting and SDS-PAGE.

Line 221-222: It says Cys3-Protein G. I believe the authors meant Cy3, without the "s".

Line 303: I suggest giving the Kd in nanomolar, since it would more compliant to the standard using for such measurements.

Line 315: The authors refer to Fig 4 when in fact, I'm led to believe, that they want to refer to figure 3.

Line 320: correct the spelling of the word "later"

Reviewer #2:

Remarks to the Author:

Guk et al. describe the development of an antibody for the detection of I223R/H275Y Influenza

HA, which commonly renders the virus resistant to antiviral drugs, including oseltamivir. Furthermore, the authors describe the incorporation of this antibody within a lateral flow immunoassay. Even though the flow of experiments is logical, I do not believe that the manuscript at its current stage justifies the conclusions. Most importantly, all tests are performed with recombinant proteins or virus. In order to lay claim that this platform, in particular in combination with the LFI, would have any impact on clinical management of influenza virus infection, it would be absolutely essential that the authors evaluate performance of their assay directly in patient samples.

Many diagnostic platforms have ultimately failed and a rigorous assessment using nasal swabs or bronchoalveolar lavage samples would be needed to assess the performance of this.

Furthermore, the authors do not discuss the relevance of other drug resistance-associated variants in Influenza virus subtypes. Even though I223R/H275Y is an important resistance-associated variant, there are others playing an important role in other influenza virus subtypes. As far as I understand from the manuscript, the here-described antibody is specific to the H1N1 strain, even though it is not clear whether the authors tested other influenza virus subtypes. This would need to be included to ascertain the diagnostic validity of the assay.

Reviewer #3:

None

Response to the Reviewers' comments:

We appreciate the reviewers for valuable comments to improve our manuscript. The changes in the manuscript and the answers to the reviewers' comments are as follows:

Reply to Reviewer 1

Reviewer comments

In this work, the authors develop a new monoclonal antibody capable of specifically recognizing an influenza neuraminidase double-mutant which confers drug resistance. The goal stated by them is to create a point-of-care fast diagnosis method for drug-resistant strains. This is a real problem identified by the authors and they go in the direction of solving it.

This is a very respectable work but, despite the paper's merits, it cannot be framed as a protein engineering paper neither as diagnostic development paper since it lacks in both areas. I find that there are major issues that have to be addressed to substantiate their claims before it can be accepted in a publication such as Nature Communications.

Question 1

They show the antibody recognizing the double-mutant protein and not recognizing the wild-type. However, they never use single-mutants in their study which also confers drug resistance and is more prevalent than the double mutants. It's paramount to test the single mutants to show the diagnostic value of the antibody.

Answer) Following the reviewer's suggestion, we tested the detection of single-mutant influenza virus using A4 antibody. Because H275Y mutation is the most frequently observed drug-resistant mutation,^[1] we examined the diagnostic ability of A4 antibody for pH1N1/H275Y mutant virus. The pH1N1/H275Y mutant virus (H275Y mutation A/Korea2785/2009 pdm: NCCP 42017) was obtained from the National Culture Collection for Pathogens (NCCP) operated by the Korea National Institute of Health (KNH).

Figure R1a shows binding activity of purified A4 antibody to H275Y NA by competition ELISA. The A4 antibody bound to H275Y NA in a concentration-dependent manner with K_d of 0.12 μ M. Figure R1b displays the interaction between A4 antibody and pH1N1/H275Y mutant virus (10^7 PFU/mL) by dot-blot analysis. A4 antibody was applied to pH1N1/H275Y mutant virus and HRP-conjugated anti-human IgG Fc was applied for detection. For the comparison, I223Y/H275Y pH1N1 (10^7 PFU/mL) and wt NA were also examined. As shown

in Figure R1b, the dot was observable only from the double-mutant virus. This suggests the low affinity of A4 antibody to the single-mutant influenza virus.

Figure R1. (A) Binding activity of purified A4 antibody to H275Y NA by competition ELISA. (B) Interaction of A4 antibody to I223Y/H275Y pH1N1 (10^7 PFU/mL), H275Y pH1N1 (10^7 PFU/mL), and wt NA (0.5 mg/mL) by dot-blot analysis.

We also applied A4 antibody for the detection of pH1N1/H275Y mutant virus by using colorimetry, SERS, and LFA. Figure R2a is absorption spectra of A4-Au NPs in the presence of pH1N1/H275Y mutant virus. The presence of single-mutant virus in the A4-Au NP solutions caused little change in absorption spectra shift. The SERS-based immunoassay result for H275Y pH1N1 (10^6 PFU) also shows negative signals (Figure R2b). Lastly, the micrograph of the LFA after detection of single-mutant virus (10^7 PFU) exhibits no test line. Taken together, we concluded that the A4 antibody can recognize the I223R/H275Y pH1N1 virus specifically.

Figure R2. (A) Absorption spectra of A4-Au NPs in the presence of H275Y pH1N1. (B) SERS spectra of MGITC obtained from NPs-on-plate structures in the presence of H275Y

pH1N1 (10^6 PFU). (C) Optical image of lateral flow system after detection of H275Y pH1N1 (10^7 PFU).

Although we reported the detection of multidrug-resistant virus in this manuscript, the identification of single-mutant influenza virus is also important as mentioned by the reviewer. To achieve this goal, we had been developed the methods for pH1N1/H275Y mutant virus.^[2-4] Moreover, we will report the other state-of-art results for the accurate identification of drug-resistant virus soon.

- [1] Baek, Y. H.; Song, M-S.; Lee, E.-Y.; Kim, Y.; Kim, E.-H.; Park, S.-J.; Park, K. J.; Kwon, H.; Pascua, P. N. Q.; Lim, G.-J.; Kim, S.; Yoon, S.-W.; Kim, M. H.; Webby, R. J.; Choi, Y.-K. *J. Virol.*, **2015**, *89*, 287.
- [2] Hwang, S. G.; Ha, K.; Guk, K.; Lee, D. K.; Eom, G.; Song, S.; Kang, T.; Park, H.; Jung, J.; Lim, E.-K. *Sci. Rep.*, **2018**, *8*, 12999.
- [3] Eom, G.; Hwang, A.; Lee, D. K.; Guk, K.; Moon, J.; Jeong, J.; Jung, J.; Kim, B.; Lim, E.-K.; Kang, T. *ACS Appl. Bio Mater.*, **2019**, *2*, 1233.
- [4] Eom, G.; Hwang, A.; Kim, H.; Yang, S.; Lee, D. K.; Song, S.; Ha, K.; Jeong, J.; Jung, J.; Lim, E.-K.; Kang, T. *ACS Sens.* **2019**, *4*, 2282.

Figure R1, R2 were added in Supplementary Information and we changed the manuscript as follows.

“The pH1N1/H275Y mutant virus (H275Y mutation A/Korea2785/2009 pdm: NCCP 42017) was obtained from the National Culture Collection for Pathogens (NCCP) operated by the Korea National Institute of Health (KNH).” (Line 12, Page 6).

“Additionally, we tested the recognition of single-mutant influenza NA protein (H275Y NA) using A4 antibody. H275Y mutation is the most frequently observed drug-resistant mutation.¹⁰ The A4 antibody bound to H275Y NA in a concentration-dependent manner with K_d of 0.12 μ M (Figure S4A). Figure S4B displays the interaction between A4 antibody and pH1N1/H275Y mutant virus (10^7 PFU/mL) by dot-blot analysis. For the comparison, I223R/H275Y pH1N1 (10^7 PFU/mL) and wt NA were also examined. As shown in Figure S4B, the dot was observable only from the double-mutant virus. This suggests the low affinity of A4 antibody to the single-mutant influenza virus.” (Line 20, Page 17).

“The presence of wt pH1N1 virus and H275Y pH1N1 virus in the A4-Au NP reaction solutions caused little change in color or absorption spectral shift (Figure 4A, S6C, S7A).” (Line 5, Page 21).

“For the detection of influenza viruses, the immune substrates were reacted with I223R/H275Y pH1N1, wt pH1N1, or H275Y pH1N1, and then the immunoprobes were reacted (Figure S8). Figure 5a is the SERS-based immunoassay results for I223R/H275Y pH1N1 (blue spectrum) and wt pH1N1 (black spectrum). The number of both viruses is 1,500 PFU. The SERS-based immunoassay result for H275Y pH1N1 was shown in Figure S7B. When the sample includes I223R/H275Y mutant virus, Au NPs on a nanoplate (NPs-on-plate) structures can be constructed through the immunoreaction of A4-I223R/H275Y pH1N1-HA. This NPs-on-plate architectures can provide significantly enhanced SERS signals. In contrast, very weak SERS signals were obtained when the sample has wt pH1N1 or H275Y pH1N1 because A4 does not bind to the wt influenza virus or single-mutant virus.” (Line 1, Page 22).

“On the other hand, wt pH1N1 virus and H275Y pH1N1 virus are not able to interact with A4-Au NPs; thus, a signal from the test line is not observed. Figure 6A, S7C is the micrographs of the LFAs after detection of wt, double-mutant, and single-mutant viruses. When the I223R/H275Y pH1N1 virus samples were applied, the red test lines were observed clearly. Importantly, even in the cases of high concentrations of wt pH1N1 virus (10^6 PFU) and H275Y virus (10^6 PFU), only the control line was observed in the absence of the I223R/H275Y pH1N1 virus.” (Line 9, Page 23).

Question 2

They frame the paper as a diagnostic development work, but all the tests are done against recombinant protein or laboratory produced viruses. It's paramount that their very interesting tools are tested on real human samples where it should be compared to traditional diagnostic methods such as DNA sequencing and/or qPCR to prove its ability to accurately differentiate the virus in real samples. Sensitivity and specificity numbers should be calculated after these tests to show the validity of the methods.

Answer) The current diagnosis in the hospital does not confirm the presence of antiviral drug-resistance of influenza viruses including I223R/H275Y pH1N1. The collected nasopharyngeal swab samples are diagnosed as influenza-positive or -negative by multiplex RT-PCR. Thus, I223R/H275Y pH1N1 positive patient samples could not be selectively

obtained from the hospital. To demonstrate whether the developed A4-Au NPs can selectively detect I223R/H275Y pH1N1 virus in real samples, influenza-positive nasopharyngeal swab samples (n = 14, Ct (threshold cycle) = 18.76 - 28.03) were mixed with I223R/H275Y pH1N1 virus (10^3 PFU). Figure R3a is optical images of A4-based lateral flow systems after detection of influenza-positive nasopharyngeal swab samples in the absence of I223R/H275Y pH1N1 virus where only the control line was observed. On the other hand, the red test lines were observed clearly of influenza-positive nasopharyngeal swab samples in the presence of I223R/H275Y pH1N1 virus (Figure R3b). Based on this result, we concluded that the A4 antibody can recognize the I223R/H275Y pH1N1 virus specifically in real sample. The sensitivity and specificity of A4-based LFA developed for rapid antiviral multidrug-resistant influenza virus diagnostic tests are 100% (14/14) and 100% (14/14), respectively.

Figure R3. Optical images of A4-based lateral flow systems after detection of influenza-positive nasopharyngeal swab samples (A) in the absence or (B) presence of I223R/H275Y pH1N1 virus (10^3 PFU).

Figure R3 was added in the revised manuscript as Figure 7 and we changed the manuscript as follows.

“Nasopharyngeal samples from patients with influenza-like symptoms were collected with the flocced nasopharyngeal swabs and placed into the virus transport media (UTM, Copan

Diagnostics Inc., USA). Each sample was analyzed with the AdvanSure RV RT-PCR kit (LG Life Sciences, Korea) following the manufacturer's instructions with the SLAN Real-Time Quantitative PCR Detection System (LG Life Sciences, Korea). A Ct (threshold cycle) value of 25 was used for the cut-off for influenza-positivity according to the manufacturer's recommendation. All samples were stored at -70 °C before use. The protocol for this retrospective study was reviewed and approved by the Institutional Review Board of Yonsei University Health Service, Severance Hospital, Seoul, Korea (IRB approval number: 4-2017-1179).” (Line 17, Page 7).

“Lastly, the LFA was examined in the influenza-positive nasopharyngeal swab samples.” (Line 2, Page 15).

“Finally, we investigated that the A4-based LFA can be used for the diagnosis of multidrug-resistant influenza virus in real human samples. Because the current diagnosis in the hospital does not confirm the presence of antiviral drug-resistance of influenza viruses, nasopharyngeal samples from patients with influenza-like symptoms were collected and mixed with I223R/H275Y pH1N1 virus (10^3 PFU). Figure 7a is optical images of A4-based lateral flow systems after detection of influenza-positive nasopharyngeal swab samples in the absence of I223R/H275Y pH1N1 virus where only the control line was observed. On the other hand, the red test lines were observed clearly after detection of influenza-positive nasopharyngeal swab samples in the presence of I223R/H275Y pH1N1 virus (Figure 7b). This verified that the A4 antibody can recognize the I223R/H275Y pH1N1 virus specifically in real sample. The sensitivity and specificity of A4-based LFA developed for rapid antiviral multidrug-resistant influenza virus diagnostic tests are 100% (14/14) and 100% (14/14), respectively.” (Line 11, Page 24).

Question 3-1

They show computational studies for docking of the antibody-antigen complex without providing any methodological detail. Free energy number is discussed but it's never explained how does the numbers were obtained. Different from protein structure modeling and despite its evolution in the past years, protein-protein docking methods are still only valid after experimental validation. Given two structures, a docking algorithm will always find many docking conformations even if the two proteins don't actually bind. Without experimental validation of the docking structure, it's impossible to assess its quality and

validity. Without further evidence, there is no way to substantiate the claims made. Moreover, all the features pointed out for increased binding affinity consider hypothetical enthalpic changes, when in fact it could be entropic contributions.

Answer) With respect to the docking simulations of the epitope in the CDR of A4, a total of 20 conformations of the epitope were generated with the genetic algorithm. Among these putative binding conformations, clustered together had similar binding modes differing by less than 1.5 Å in positional root-mean-square deviation. The lowest-energy configuration in the top-ranked cluster was selected as the final structural models for the antibody-epitope complexes. To explain these, we have added a paragraph in the revised manuscript as follows.

“*Docking simulations of wt and I223R/H275Y NA in the CDR of A4*

3D structure of A4 obtained in the precedent homology modeling served as the receptor model in docking simulations with wt and I223R/H275Y mutant NA. The epitope structures were extracted from the X-ray crystal structure of (PDB entry: 4B7R).³⁷ Docking simulations were carried out using the AutoDock program to estimate the binding free energy (ΔG_{bind}) of the epitope in the complementarity-determining region (CDR) of A4, which can be expressed mathematically as follows.³⁸

$$\begin{aligned} \Delta G_{bind} = & W_{vdW} \sum_i \sum_{j>i} \left(\frac{A_{ij}}{r_{ij}^{12}} - \frac{B_{ij}}{r_{ij}^6} \right) + W_{hbond} \sum_i \sum_{j>i} E(t) \left(\frac{C_{ij}}{r_{ij}^{12}} - \frac{D_{ij}}{r_{ij}^{10}} \right) \\ & + W_{elec} \sum_i \sum_{j>i} \frac{q_i q_j}{\epsilon(r_{ij}) r_{ij}} + W_{tor} N_{tor} + W_{sol} \sum_i^{atoms} S_i \left(O_i^{max} - \sum_{j \neq i}^{atoms} V_j e^{-\frac{r_{ij}^2}{2\sigma^2}} \right) \end{aligned} \quad (1)$$

The weighting parameters for van der Waals contacts (W_{vdW}), hydrogen bonds (W_{hbond}), electrostatic interactions (W_{elec}), entropic penalty (W_{tor}), and ligand dehydration free energy (W_{sol}) were set to 0.1485, 0.0656, 0.1146, 0.3113, and 0.1711, respectively, as in the original AutoDock program. r_{ij} stands for the interatomic distance, and A_{ij} , B_{ij} , C_{ij} , and D_{ij} are associated with the well depth and the equilibrium distance in the potential energy function. The hydrogen bond term has the additional weighting factor ($E(t)$) to describe the angle-dependent directionality. To compute the electrostatic interaction energy between A4 antibody and the epitopes, we used the sigmoidal function with respect to r_{ij} proposed by Mehler *et al.* as the distance-dependent dielectric constant.³⁹ In the entropic penalty term, N_{tor} indicates the number of rotatable bonds in the epitope. In the hydration free energy term, S_i and V_i denote the atomic solvation energy per unit volume and the fragmental atomic volume, respectively, while Occ_i^{max} represents the maximum occupancy of each atom in the epitope.⁴⁰

All the energy parameters in Eq. (1) were extracted from the original AutoDock program to derive the binding modes of wt and I223R/H275Y mutant NA in the CDR of A4.

Among 20 conformations generated with the genetic algorithm, those clustered together had similar binding modes differing by less than 1.5 Å in positional root-mean-square deviation. The lowest-energy configuration in the top-ranked cluster was selected as the final structural models for antigen-antibody complexes.” (Line 22, Page 11).

Question 3-2

They show computational studies for docking of the antibody-antigen complex without providing any methodological detail. Free energy number is discussed but it's never explained how does the numbers were obtained. Different from protein structure modeling and despite its evolution in the past years, protein-protein docking methods are still only valid after experimental validation. Given two structures, a docking algorithm will always find many docking conformations even if the two proteins don't actually bind. Without experimental validation of the docking structure, it's impossible to assess its quality and validity. Without further evidence, there is no way to substantiate the claims made. Moreover, all the features pointed out for increased binding affinity consider hypothetical enthalpic changes, when in fact it could be entropic contributions.

Answer) We agreed that the binding modes derived from docking simulations had to be validated with experimental approaches. Therefore, we carried out the mutational analysis at positions His94 in the light chain and Trp33 in the heavy chain in order to assess the importance of the hydrophobic interactions to stabilize the epitopes in the CDR of A4. These mutant A4 antibodies were purified with the same method as A4 antibody. Figures R4 show the binding affinities of I223R/H275Y NA with respect to the wt and the mutant A4 antibodies. We note that the mutation of A4 antibody at position 94 in the light chain from His to Ala leads to approximately 50-fold increase in the K_d value associated with binding of I223R/H275Y NA (183 nM). This indicates the significant role of His94 in the light chain for the stabilization of the epitope. Similarly, the K_d value of I223R/H275Y NA increases from 3.50 to 150 nM in going from the wt to the W33A mutant in the heavy chain of A4.

Figure R4. (A) Binding activity of H94A mutant A4 antibody to I223R/H275Y NA by competition ELISA. (B) Binding activity of W33A mutant A4 antibody to I223R/H275Y NA by competition ELISA.

The results of mutational analyses are consistent with those of docking simulations indicating that the strengthening of hydrophobic interactions with aromatic sidechains is responsible for the tight binding of I223R/H275Y NA in the CDR of A4. Judging from the consistency between the experimental and computational results, the capability of forming a van der Waals contact with the aromatic residues in CDR seems to be a determinant for selective binding to A4 antibody. To present and discuss the newly found experimental results, we added Figure R4 in Supplementary Information and the paragraph in the revised manuscript as follows.

“Validation of docking simulation results with mutational analysis

In order to assess the importance of the hydrophobic interactions to stabilize the epitopes in the CDR of A4, we carried out the mutational analysis at positions His94 in the light chain and Trp33 in the heavy chain. These mutant A4 antibodies were purified with the same method as the wild types. Figures S5 shows the binding affinities of I223R/H275Y NA with respect to the two kinds of mutant A4 antibodies (H94A and W33A mutant A4 antibodies). We note that the mutation of A4 antibody at position 94 in the light chain from His to Ala leads to approximately 50-fold increase in the K_d value associated with binding of I223R/H275Y NA (183 nM). This indicates the significant role of His94 in the light chain in the stabilization of the epitope. Similarly, the K_d value of I223R/H275Y NA increases from 3.50 to 150 nM in going from the wt to the W33A mutant in the heavy chain. The results of mutational analyses are thus consistent with those of docking simulations indicating that the strengthening of hydrophobic interactions with aromatic sidechains are responsible for tight binding of I223R/H275Y NA in the CDR of A4. Judging from the consistency between the

experimental and computational results, the capability of forming a van der Waals contact with the aromatic residues in CDR seems to be a determinant for selective binding to A4 antibody.” (Line 7, Page 19).

Question 3-3

They show computational studies for docking of the antibody-antigen complex without providing any methodological detail. Free energy number is discussed but it's never explained how does the numbers were obtained. Different from protein structure modeling and despite its evolution in the past years, protein-protein docking methods are still only valid after experimental validation. Given two structures, a docking algorithm will always find many docking conformations even if the two proteins don't actually bind. Without experimental validation of the docking structure, it's impossible to assess its quality and validity. Without further evidence, there is no way to substantiate the claims made. **Moreover, all the features pointed out for increased binding affinity consider hypothetical enthalpic changes, when in fact it could be entropic contributions.**

Answer) The binding free energy function used in this work included not only the enthalpic term but also the entropic term that is proportional to the number of rotatable bonds in the epitope. To place an emphasis on this point, we added a sentence in the revised manuscript as “In the entropic penalty term, N_{tor} indicates the number of rotatable bonds in the epitope. In the hydration free energy term, S_i and V_i denote the atomic solvation energy per unit volume and the fragmental atomic volume, respectively, while Occ_i^{max} represents the maximum occupancy of each atom in the epitope.”⁴⁰ (Line 13, Page 12).

Question 4

There are no considerations or references on phage display library specifications. This is very important for an antibody display paper.

Answer) Previously constructed large naïve human Fab phage display library (3×10^{10}) in Korea Research Institute of Bioscience and Biotechnology was used for antibody screening.

[1] Kim, S.; Park, I.; Park, S. G.; Cho, S.; Kim, J. H.; Ipper, N. S.; Choi, S. S.; Lee, E. S.; Hong, H. J. *Molecules and Cells*, **2017**, *40*, 656.

We added the description in the revised manuscript as “For antibody screening, previously constructed large naïve human antigen-binding fragment (Fab) phage display library (3×10^{10}) in Korea Research Institute of Bioscience and Biotechnology was used.³¹” (Line 21, Page 8).

Additional reviewer comments

Those are the major points that I firmly believe should be addressed to make this paper, that has a lot of potentials, a very important piece of work that will push the current state-of-the-art on influenza diagnosis. Other minor points are described below.

Question 5

Affinities measured by ELISA are used for a rough estimation. Since the authors measured the affinity of the antibody by surface plasmon resonance, which is the state-of-art technique for such, the ELISA measurements should be kept as supplemental information or omitted since it does not add anything to the paper. Moreover, the methodology for measuring the affinity by ELISA was not described in the material and methods section.

Answer) ELISA results were added in Supplementary Information. Affinity measurements were performed by competitive ELISA as follows. Microtiter wells were coated with the purified NA (100 ng) in 50 mM sodium carbonate buffer (pH 9.6) at 4 °C overnight, blocked with BSA (2%) in PBS, and washed with PBST. A reaction mixture containing purified antibody (10 nM) and various concentrations (10^{-11} - 10^{-5} M) of NA as a competing antigen were pre-incubated at 37 °C for 1 - 2 h. The mixture was then added to each well previously coated with 100 ng of NA. Anti-human Fc-HRP (Thermo, 1:10,000 v/v) was added to the wells. All incubations were carried out at 37 °C for 1 h. Color was developed with OptEIA TMB Substrate (BD), and the absorbance was measured at 450 nm in a microtiter plate reader. Affinity was determined as the antigen concentration required to inhibit 50% of binding activity and K_d value was calculated from a Klotz plot.

We modified the manuscript as “Microtiter wells were coated with the purified NA (100 ng) in 50 mM sodium carbonate buffer (pH 9.6) at 4 °C overnight, blocked with BSA (2%) in PBS, and washed with PBST. A reaction mixture containing purified antibody (10 nM) and various concentrations (10^{-11} - 10^{-5} M) of NA as a competing antigen were pre-incubated at 37 °C for 1 - 2 h. The mixture was then added to each well previously coated with 100 ng of NA. HRP - conjugated goat anti-human IgG (Pierce) was used for the detection of bound IgG. Color was developed with the 3,3',5,5'-Tetramethylbenzidine substrate reagent set (BD Biosciences), and the absorbance at 450 nm was measured using a microtiter plate reader (Emax; Molecular

Devices). Affinity was determined as the antigen concentration required to inhibit 50% of binding activity and binding affinity (K_d) value was calculated from a Klotz plot.” (Line 14, Page 10).

Question 6

There is no methodology for docking described in the paper and also there is no methodology for free energy calculation described in the paper. Please clarify this.

Answer) Please, refer the answer of Question 3-1.

Question 7

Different from protein structure modelling and despite its evolution in the past years, protein-protein docking methods are still only valid after experimental validation. Given two structures, a docking algorithm will always find many docking conformations even if the two proteins don't actually bind. Without experimental validation of the docking structure, it's impossible to assess its quality and validity. Without further evidence there is no way to substantiate the claims made. Moreover, all the features pointed out for increased binding affinity consider hypothetical enthalpic changes, when in fact it could be entropic contributions.

Answer) Please, refer the answer of Question 3-2 and 3-3.

Question 8

Line 34: It abbreviates neuraminidase to “NA” which has never been used in the text and so far.

Answer) We changed the NA to neuraminidase in the revised manuscript.

Question 9

Line 121: please correct Delbecco to Dulbecco.

Answer) We changed the Delbecco to Dulbecco in the revised manuscript.

Question 10

Line 130: why did you used insect expression system. Instead of mammalian expression system?

Answer) Previously, soluble NA protein from the 1918 H1N1 (A/Brevig Mission/1/1918) strain was successfully expressed using a baculovirus expression system and crystalized for structural analysis.^[1] Therefore, we used insect expression system.

[1] Xu, X.; Zhu, X.; Dwek, R. A.; Stevens, J.; Wilson, I. A. *J. Virol.*, **2008**, 82, 10493.

Question 11

Line 138: the centrifugation for 1h at 16,000g was to precipitate the protein or cell debris? I assume the protein was in the supernatant and that was used for affinity chromatography purification, but the text suggests that the “protein was protein was obtained by centrifugation” which implies that the protein was pelleted.

Answer) We agree with the reviewer’s comment. The corresponding sentence in the manuscript has been changed as “After the cell lysate was sonicated to reduce its viscosity, the cell debris was removed by centrifugation for 1 h at 16,000 g. The soluble protein from the cell supernatant was applied to Ni-Nitrilotriacetic acid agarose resin (Qiagen), washed, and eluted with buffer (50 mM Tris-HCl, 0.5 M NaCl, 0.5 M imidazole, pH 8.0).” (Line 13, Page 8).

Question 12

Line 146: where the antibody library comes from? Is there a reference for this library? Is it human? Is it scFv or Fab format? What is the promoter? What is the helper phage used?

Answer) Previously constructed large naïve human Fab library (3×10^{10}) in Korea Research Institute of Bioscience and Biotechnology was used for antibody screening.^[1] The Fab display vector contains a bicistronic operon under the control of *LacZ* promoter. VCSM13 helper phages was used.^[2]

[1] Kim, S.; Park, I.; Park, S. G.; Cho, S.; Kim, J. H.; Ipper, N. S.; Choi, S. S.; Lee, E. S.; Hong, H. J. *Molecules and Cells*, **2017**, *40*, 656.

[2] Zhu, Z.; Dimitrov, D. S. *Methods Mol. Biol.*, **2009**, *525*, 129.

We added the description in the revised manuscript as “For antibody screening, previously constructed large naïve human antigen-binding fragment (Fab) phage display library (3×10^{10}) in Korea Research Institute of Bioscience and Biotechnology was used.³¹ A phagemid vector (KRIBB-Fab) contains a bicistronic operon under the control of *LacZ* promoter.” (Line 21, Page 8).

Question 13

Line 147: How many washes were performed in each round? It’s very important for the authors to give details of the method used to obtain the antibody so others can successfully follow.

Answer) Following the suggestion, we have added the number of washes performed in each round.

“Four rounds of panning were conducted, and the stringency of selection was increased with each round by gradually increasing the number of washes from 10 to 40.” (Line 6, Page 9).

Question 14

Line 156: How was the soluble Fab expressed? Was it fused to g3p? Is there an amber stop codon between the Fab gene and g3p so one can express soluble Fab when using a non-suppressor strain?

Answer) Soluble Fab expression was induced in *E. coli* TG1 cells at 30°C overnight by adding isopropyl β -D-1-thiogalactopyranoside to a final concentration of 1 mM.

We added the description in the revised manuscript as “To screen individual clones for specific binding to I223R/H275Y NA, 500 colonies were randomly selected from the output plate after the third or fourth round of panning, cultured in Superbroth medium containing 100 μ g/mL ampicillin until optical density of 0.5, and induced for Fab expression in *Escherichia coli* TG1 cells at 30 °C overnight by adding isopropyl β -D-1-thiogalactopyranoside to a final concentration of 1 mM.” (Line 9, Page 9).

Question 15

Line 157: how was the Fab detected? Is there a tag so one can use a labeled anti-tag antibody? Please give more details.

Answer) A microtiter plate was coated with 100 ng of I223R/H275Y NA in coating buffer (0.5 M carbonate buffer, pH 9.6) and incubated at 4 °C overnight. After blocking, Goat F(ab')₂ Anti-Human IgG (Fab')₂-HRP (Abcam) antibody was used for the colorimetric detection of bound clones using the tetramethylbenzimidine substrate.

We added the description in the revised manuscript as “In detail, a microtiter plate was coated with 100 ng of I223R/H275Y NA in coating buffer (0.05 M carbonate buffer, pH 9.6) and incubated at 4 °C overnight. After blocking, Goat F(ab')₂ Anti-Human IgG (Fab')₂-HRP (Abcam) antibody was used for the colorimetric detection of bound clones using the tetramethylbenzimidine substrate.” (Line 15, Page 9).

Question 16

Line 160: Did the library only contain kappa light chains and no lambda light chains?

Answer) Yes, this library was designed to include only kappa chains.

Question 17

Line 164: What polymerase was used? Please insert in the text.

Answer) Pfu DNA Polymerase (Thermo Scientific) was used and added to the text.

Question 18

Line 165: Is recombinant PCR the same as overlap extension PCR? If so, I believe the later term is more widely used and well understood.

Answer) Following the suggestion of the reviewer, we switched to “overlap extension PCR”.

Question 19

Line 166: The fragments were not subcloned, but merely cloned. Subcloning is the technique where a fragment is directly excised with restriction enzyme from one plasmid and the ligated into another. In this case, the fragments were amplified by PCR, modified and then cloned.

Answer) According to reviewer’s suggestion “subcloned” has been replaced by “cloned”.

Question 20

Line 176: What was the Bioanalyzer used for? It’s not clear. It looks like it was used for western blotting and SDS-PAGE.

Answer) Bioanalyzer was not used for western blotting or SDS-PAGE and was mistakenly inserted. Therefore, it has been removed from the text.

Question 21

Line 221-222: It says Cys3-Protein G. I believe the authors meant Cy3, without the “s”.

Answer) Cys3-protein G is the three cysteine-tagged protein G. We added the explanation about Cys3-protein G in the revised manuscript.

Question 22

Line 303: I suggest giving the Kd in nanomolar, since it would more compliant to the standard using for such measurements.

Answer) We changed the manuscript as “steady state affinity (K_D) was determined to be 0.254 nM.” (Line 14, Page 17).

Question 23

Line 315: The authors refer to Fig 4 when in fact, I’m led to believe, that they want to refer to figure 3.

Answer) We changed the typo.

Question 24

Line 320: correct the spelling of the word “later”.

Answer) The word “latter” in this sentence is correct. It indicates the A4-wt NA complex.

Replies to Reviewer 2

Reviewer comments

Guk et al. describe the development of an antibody for the detection of I223R/H275Y Influenza HA, which commonly renders the virus resistant to antiviral drugs, including oseltamivir. Furthermore, the authors describe the incorporation of this antibody within a lateral flow immunoassay. Even though the flow of experiments is logical, I do not believe that the manuscript at its current stage justifies the conclusions.

Question 1

Most importantly, all tests are performed with recombinant proteins or virus. In order to lay claim that this platform, in particular in combination with the LFI, would have any impact on clinical management of influenza virus infection, it would be absolutely essential that the authors evaluate performance of their assay directly in patient samples. Many diagnostic platforms have ultimately failed and a rigorous assessment using nasal swabs or bronchoalveolar lavage samples would be needed to assess the performance of this.

Answer) The current diagnosis in the hospital does not confirm the presence of antiviral drug-resistance of influenza viruses including I223R/H275Y pH1N1. The collected nasopharyngeal swab samples are diagnosed as influenza-positive or -negative by multiplex RT-PCR. Thus, I223R/H275Y pH1N1 positive patient samples could not be selectively obtained from the hospital. To demonstrate whether the developed A4-Au NPs can selectively detect I223R/H275Y pH1N1 virus in real samples, influenza-positive nasopharyngeal swab samples (n = 14, Ct (threshold cycle) = 18.76 - 28.03) were mixed with I223R/H275Y pH1N1 virus (10^3 PFU). Figure R3a is optical images of A4-based lateral flow systems after detection of influenza-positive nasopharyngeal swab samples in the absence of I223R/H275Y pH1N1 virus where only the control line was observed. On the other hand, the red test lines were observed clearly of influenza-positive nasopharyngeal swab samples in the presence of I223R/H275Y pH1N1 virus (Figure R3b). Based on this result, we concluded that the A4 antibody can recognize the I223R/H275Y pH1N1 virus specifically in real sample. The sensitivity and specificity of A4-based LFA developed for rapid antiviral multidrug-resistant influenza virus diagnostic tests are 100% (14/14) and 100% (14/14), respectively.

Figure R3. Optical images of A4-based lateral flow systems after detection of influenza-positive nasopharyngeal swab samples (A) in the absence or (B) presence of I223R/H275Y pH1N1 virus (10^3 PFU).

Figure R3 was added in the revised manuscript as Figure 7 and we changed the manuscript as follows.

“Nasopharyngeal samples from patients with influenza-like symptoms were collected with the flocked nasopharyngeal swabs and placed into the virus transport media (UTM, Copan Diagnostics Inc., USA). Each sample was analyzed with the AdvanSure RV RT-PCR kit (LG Life Sciences, Korea) following the manufacturer’s instructions with the SLAN Real-Time Quantitative PCR Detection System (LG Life Sciences, Korea). A Ct (threshold cycle) value of 25 was used for the cut-off for influenza-positivity according to the manufacturer’s recommendation. All samples were stored at $-70\text{ }^{\circ}\text{C}$ before use. The protocol for this retrospective study was reviewed and approved by the Institutional Review Board of Yonsei University Health Service, Severance Hospital, Seoul, Korea (IRB approval number: 4-2017-1179).” (Line 17, Page 7).

“Lastly, the LFA was examined in the influenza-positive nasopharyngeal swab samples.” (Line 2, Page 15).

“Finally, we investigated that the A4-based LFA can be used for the diagnosis of multidrug-resistant influenza virus in real human samples. Because the current diagnosis in the hospital does not confirm the presence of antiviral drug-resistance of influenza viruses, nasopharyngeal samples from patients with influenza-like symptoms were collected and mixed with I223R/H275Y pH1N1 virus (10^3 PFU). Figure 7a is optical images of A4-based lateral flow systems after detection of influenza-positive nasopharyngeal swab samples in the absence of I223R/H275Y pH1N1 virus where only the control line was observed. On the other hand, the red test lines were observed clearly after detection of influenza-positive nasopharyngeal swab samples in the presence of I223R/H275Y pH1N1 virus (Figure 7b). This verified that the A4 antibody can recognize the I223R/H275Y pH1N1 virus specifically in real sample. The sensitivity and specificity of A4-based LFA developed for rapid antiviral multidrug-resistant influenza virus diagnostic tests are 100% (14/14) and 100% (14/14), respectively.” (Line 11, Page 24).

Question 2

Furthermore, the authors do not discuss the relevance of other drug resistance-associated variants in Influenza virus subtypes. Even though I223R/H275Y is an important resistance-associated variant, there are others playing an important role in other influenza virus subtypes. As far as I understand from the manuscript, the here-described antibody is specific to the H1N1 strain, even though it is not clear whether the authors tested other influenza virus subtypes. This would need to be included to ascertain the diagnostic validity of the assay.

Answer) We tested the present mutant virus sensing methods by using four kinds of influenza virus subtypes. The reverse genetics system was used to generate different subtypes of the I223R/H275Y influenza virus. The expression plasmids for the eight-plasmid reverse genetic system were kindly donated by Dr. Meehyein Kim of Korea Research Institute of Chemical Technology (Daejeon, Republic of Korea). NA and HA genes derived from A/Puerto Rico/8/1934 (H1N1), A/Brisbane/10/2007 (H3N2), A/swine/Korea/GC0503/2005 (H1N1), and A/canine/Korea/MV1/2012 (H3N2) genomic RNAs were cloned individually into pVP-NA or pVP-HA vector^[1] using universal reverse primers and genome-specific primers.^[2] In addition, I223R/H275Y mutation was introduced within the NA fragment by the forward primer: 5'-TATTCGTCTCAGGGATGAAGACTATCATTGCTTTGAGCTACATT-3' and reverse primer: 5'-ATATCGTCTCGTATTTGTTTTTAATTAATGCACTCAAATGCAAA-3'. Briefly, co-cultured 293T and MDCK cells (0.5×10^6 cells per well) grown in six-well plate were transfected using Lipofectamine 3000 (Thermo Fisher Scientific) according to the

manufacturer's instructions (1 μg of each influenza DNA plasmid). After 6 h of incubation, the transfection medium was replaced by Opti-MEM. After 30 h of transfection, each well was supplemented with 1 ml of Opti-MEM containing 0.75 $\mu\text{g}/\text{ml}$ tosylsulfonyl phenylalanyl chloromethyl ketone-trypsin. At 3 to 6 days post-transfection, cell supernatants were titrated onto MDCK cell monolayers to estimate influenza virus titers. All experiments were done in triplicate.

To examine that the developed A4-Au NPs can detect various subtypes of I223R/H275Y influenza virus in real samples, influenza-positive nasopharyngeal swab samples ($n = 6$, $\text{Ct} = 18.76 - 28.03$) were mixed with 10^3 PFU of I223R/H275Y influenza viruses (A/Puerto Rico/8/1934 (H1N1), A/Brisbane/10/2007 (H3N2), A/swine/Korea/GC0503/2005 (H1N1), and A/canine/Korea/MV1/2012 (H3N2)). Figure R5 is optical image of A4-based lateral flow system after detection of influenza-positive nasopharyngeal swab samples in the absence or presence of I223R/H275Y influenza viruses, in which positive test lines were clearly observed in influenza-positive nasopharyngeal swab samples taken with various subtypes of I223R/H275Y influenza virus. Based on this result, we concluded that the A4 antibody can recognize the various subtypes of I223R/H275Y influenza virus specifically in real sample.

Figure R5. Optical images of A4-based lateral flow systems after detection of influenza-positive nasopharyngeal swab samples in the absence or presence of I223R/H275Y pH1N1 influenza viruses (A/Puerto Rico/8/1934 (H1N1), A/Brisbane/10/2007 (H3N2), A/swine/Korea/GC0503/2005 (H1N1), and A/canine/Korea/MV1/2012 (H3N2)).

- [1] Kim, M.; Kim, S. Y.; Lee, H. W.; Shin, J. S.; Kim, P.; Jung, Y. S.; Jeong, H. S.; Hyun, J. K.; Lee, C. K. *Antiviral Research*, **2013**, *100*, 460.
- [2] Hoffman, E.; Neumann, G.; Kawaoka, Y.; Hobom, G.; Webster, R. G. *Proc. Natl. Acad. Sci. U. S. A.*, **2000**, *97*, 6108.

The representative Figure R5A was included in the revised manuscript as Figure 8 and the others were added in Supplementary Information. Also, we modified the manuscript as follows.

“Influenza virus strains including A/California/07/2009 (H1N1), A/Puerto Rico/8/1934 (H1N1), and A/Brisbane/10/2007 (H3N2) were originally provided from the Korea Centers for Disease Control and Prevention, Korea. A/swine/Korea/GC0503/2005 (H1N1) and A/canine/Korea/MV1/2012 (H3N2) were kindly provided by Prof. Dae-Sub Song (Korea University).” (Line 8, Page 6).

“The reverse genetics system was used to generate different subtypes of the I223R/H275Y influenza virus. The expression plasmids for the eight-plasmid reverse genetic system were kindly donated by Dr. Meehyein Kim of Korea Research Institute of Chemical Technology (Daejeon, Republic of Korea).²⁹⁻³⁰ NA and HA genes derived from A/Puerto Rico/8/1934 (H1N1), A/Brisbane/10/2007 (H3N2), A/swine/Korea/GC0503/2005 (H1N1), and A/canine/Korea/MV1/2012 (H3N2) genomic RNAs were cloned individually into pVP-NA or pVP-HA vector using universal reverse primers and genome-specific primers. In addition, I223R/H275Y mutation was introduced within the NA fragment by the forward primer: 5'-TATTCGTCTCAGGGATGAAGACTATCATTGCTTTGAGCTACATT-3' and reverse primer: 5'-ATATCGTCTCGTATTTGTTTTTAATTAATGCACTCAAATGCAAAA-3'. Briefly, co-cultured 293T and MDCK cells (0.5×10^6 cells per well) grown in six-well plate were transfected using Lipofectamine 3000 (Thermo Fisher Scientific) according to the manufacturer's instructions (1 μ g of each influenza DNA plasmid). After 6 h of incubation, the transfection medium was replaced by Opti-MEM. After 30 h of transfection, each well

was supplemented with 1 ml of Opti-MEM containing 0.75 µg/ml tosylsulfonyl phenylalanyl chloromethyl ketone-trypsin. At 3 to 6 days post-transfection, cell supernatants were titrated onto MDCK cell monolayers to estimate influenza virus titers. All experiments were done in triplicate.” (Line 23, Page 6).

“We also tested the present mutant virus sensing methods by using four kinds of influenza virus subtypes (A/Puerto Rico/8/1934 (H1N1), A/Brisbane/10/2007 (H3N2), A/swine/Korea/GC0503/2005 (H1N1), and A/canine/Korea/MV1/2012 (H3N2)). The virus samples were prepared by mixing each of I223R/H275Y influenza viruses (10^3 PFU) with influenza-positive nasopharyngeal swab samples (n = 6). Figure 8 and S11 are optical images of A4-based lateral flow system after detection of influenza-positive nasopharyngeal swab samples in the absence or presence of I223R/H275Y influenza viruses, in which positive test lines were clearly observed in influenza-positive nasopharyngeal swab samples taken with various subtypes of I223R/H275Y influenza virus. Based on this result, we anticipate that the A4-based LFA can be used for the rapid antiviral multidrug-resistant influenza virus diagnostic test in near future.” (Line 22, Page 23).

Reviewers' Comments:

Reviewer #1:

Remarks to the Author:

Dear Editors & Authors;

I welcome the possibility of reviewing the paper for the 2nd time. In the first version the authors developed a new antibody aiming to improve Influenza diagnosis, however, many things remained to be clarified.

In this 2nd revision, the authors have addressed the major concerns I had previously:

- They have tested the antibody against the single mutant;
- They tested the antibody against donor samples;
- They provided some experimental validation on the antibody-antigen binding mechanism;
- The methodology section has been significantly improved, providing details regarding the phage display methodology and the docking methodology.

I believe the paper has scientific merit and recommend it to be published at Nature Communications.

Best wishes,

André

Reviewer #4:

Remarks to the Author:

Review of Author's response to reviewer #2 questions.

Reviewer #2

Question #1.

Reviewer#2 raised a very valid concern on the application of the LFA in a clinical setting without vigorous validation of the assay performance using real clinical material. The authors responded by including a small study (n=14) using NP/Op swabs spiked with 10³ pfu pH1N1 viruses with I223R/H275Y markers. A few issues remain:

1). Sensitivity is often an issue for LFA assays. the authors did not demonstrate a dose effect with the detection of the I223R/H275Y in NP/OP swabs. 2). Matrix effect from different NP/OP swabs/bronchoalveolar lavage samples may also impact the sensitivity of the LFA detection, a larger number of the actual clinical samples collected from different patients are needed in order to evaluate the performance of this assay. 3). A more vigorous study using actual clinical specimen that are conducted side-by-side with sequencing are indeed needed to truly demonstrate the sensitivity and specificity of the assay and its true value in the actual clinical setting.

Question 2

"Furthermore, the authors do not discuss the relevance of other drug resistance-associated variants in Influenza virus subtypes. Even though I223R/H275Y is an important resistance-associated

variant, there are others playing an important role in other influenza virus subtypes. As far as I understand from the manuscript, the here-described antibody is specific to the H1N1 strain, even though it is not clear whether the authors tested other influenza virus subtypes. This would need to be included to ascertain the diagnostic validity of the assay. "

Reviewer #2 raised a good point. The authors did not sufficiently address this question. I223R/H275Y is a recognized marker for A(H1N1) oseltamivir/zanamivir resistance, but not necessarily for other subtypes. Furthermore, there are several other genetic markers that were identified to be associated for drug resistance, and not all drug resistance are based on neuraminidase. The authors should consider modifying the language in manuscript to address this

limitation, including in the title to avoid overstatement of the study findings, for example the title can be modified to:

“Development of novel A4 antibody for detection of neuraminidase I223R/H275Y associated antiviral multidrug-resistant influenza viruse”

We appreciate the reviewers for valuable comments to improve our manuscript. The changes in the manuscript and the answers to the reviewers' comments are as follows:

Reply to Reviewer 1

Reviewer comments

I welcome the possibility of reviewing the paper for the 2nd time. In the first version the authors developed a new antibody aiming to improve Influenza diagnosis, however, many things remained to be clarified.

In this 2nd revision, the authors have addressed the major concerns I had previously:

- They have tested the antibody against the single mutant;
- They tested the antibody against donor samples;
- They provided some experimental validation on the antibody-antigen binding mechanism;
- The methodology section has been significantly improved, providing details regarding the phage display methodology and the docking methodology.

I believe the paper has scientific merit and recommend it to be published at Nature Communications.

Best wishes,

André

Answer) Thank you for the recommendation of our manuscript in *Nature Communications*.

We appreciate for your valuable comments to improve our manuscript.

Reply to Reviewer 2

Reviewer comments

Reviewer#2 raised a very valid concern on the application of the LFA in a clinical setting without vigorous validation of the assay performance using real clinical material. The authors responded by including a small study (n=14) using NP/Op swabs spiked with 10^3 pfu pH1N1 viruses with I223R/H275Y markers. A few issues remain:

Question 1

Sensitivity is often an issue for LFA assays. the authors did not demonstrate a dose effect with the detection of the I223R/H275Y in NP/OP swabs.

Answer) Following the reviewer’s suggestion, the dose effect of I223R/H275Y detection in nasopharyngeal swab samples was demonstrated. Figure R1 shows the results of LFA-based I223R/H275Y detection by varying the concentration of mutant virus in human nasopharyngeal swab samples. The results show that A4-based LFA has high detection specificity depending on the dose of I223R/H275Y pH1N1 virus.

Figure R1. Optical images of lateral flow systems after detection of I223R/H275Y pH1N1 in nasopharyngeal swab and control samples.

Figure R1 was included in Supplementary Information and the manuscript was modified as “In addition, the does effect of I223R/H275Y detection in nasopharyngeal swab samples was demonstrated (Figure S11).” (Line 19, Page 23).

Question 2

Matrix effect from different NP/OP swabs/bronchoalveolar lavage samples may also impact the sensitivity of the LFA detection, a larger number of the actual clinical samples collected from different patients are needed in order to evaluate the performance of this assay.

Answer)

Figure R2. Optical images of A4-based lateral flow systems after detection of nasopharyngeal swab samples (A,C) in the presence or (B,D) absence of I223R/H275Y pH1N1 virus (10^3 PFU).

In addition to the previous data, an additional 26 nasopharyngeal swab samples were examined using the developed LFA, as the reviewer suggested. Figure R2 shows the results of a newly tested LFA after detecting the mutant virus in human nasopharyngeal swab samples. The test line was only observed in the presence of the I223/H275Y pH1N1 viruses. In this study, a total of 40 nasopharyngeal swab samples were tested. According to the guideline of National Institute of Food and Drug Safety Evaluation of Korea, a minimum of 20 sample results are required for approval of an *in vitro* diagnostic system. Therefore, the current results are suitable for assessing the performance of A4 antibody-based LFA assay.

The figure was included in Supplementary Information and the manuscript was changed as “Totally, we tested 40 human nasopharyngeal swab samples (Figure S12). The sensitivity and specificity of A4-based LFA developed for rapid antiviral multidrug-resistant influenza virus diagnostic tests are 100% (40/40) and 100% (40/40), respectively. According to the guideline of National Institute of Food and Drug Safety Evaluation of Korea, a minimum of 20 sample results are required for approval of an *in vitro* diagnostic system.” (Line 21, Page 24).

Question 3

A more vigorous study using actual clinical specimen that are conducted side-by-side with sequencing are indeed needed to truly demonstrate the sensitivity and specificity of the assay and its true value in the actual clinical setting.

Answer) Following the reviewer’s suggestion, sequencing tests were performed and compared to the LFA assay results. The viral sequence was identical in all samples tested as shown in Figure R3. The sensitivity and specificity of A4-based LFA assay for antiviral multidrug-resistant influenza virus diagnosis is determined as 100% (26/26) and 100% (26/26), respectively (Figure R2).

Figure R3. Sequence analysis of nasopharyngeal swab samples in the presence of I223R/H275Y pH1N1 virus, for which the mutations have been identified, is indicated by a red circle.

Question 4

(Previous Question) Furthermore, the authors do not discuss the relevance of other drug resistance-associated variants in Influenza virus subtypes. Even though I223R/H275Y is an important resistance associated variant, there are others playing an important role in other influenza virus subtypes. As far as I understand from the manuscript, the here-described antibody is specific to the H1N1 strain, even though it is not clear whether the authors tested other influenza virus subtypes. This would need to be included to ascertain the diagnostic validity of the assay.

Reviewer #2 raised a good point. The authors did not sufficiently address this question. I223R/H275Y is a recognized marker for A(H1N1) oseltamivir/zanamivir resistance, but not necessarily for other subtypes. Furthermore, there are several other genetic markers that were identified to be associated for drug resistance, and not all drug resistance are based on neuraminidase. The authors should consider modifying the language in manuscript to address this limitation, including in the title to avoid overstatement of the study findings, for example the title can be modified to: “Development of novel A4 antibody for detection of neuraminidase I223R/H275Y associated antiviral multidrug-resistant influenza virus”

Answer) The goal of this study is to find an antibody to the I223R/H275Y mutant virus, an antiviral multidrug-resistant virus for both zanamivir and oseltamivir. After careful evaluation of A4 antibody, we were successful in detecting the mutant virus in several ways. However, as the reviewer pointed out, there are certainly other genetic markers of drug resistance and not all drug resistance is based on NA. In agreement with the reviewer’s suggestion, the title of the manuscript has been modified to limit the current approach to the detection of neuraminidase I223R/H275Y associated antiviral multidrug-resistant influenza virus only.

Reviewers' Comments:

Reviewer #4:

Remarks to the Author:

In this revision, the authors provided additional data and addressed most of my comments.